# Comprehensive Evaluation and Analysis for NSFW Concept Erasure in Text-to-Image Diffusion Models

**Die Chen**[*]
East China Normal University
dchen@stu.ecnu.edu.cn

**Zhiwen Li**[*]
East China Normal University
zhiwenli@stu.ecnu.edu.cn

**Cen Chen**[†]
East China Normal University
cenchen@dase.ecnu.edu.cn

**Yuexiang Xie**
Alibaba Group
yuexiang.xyx@alibaba-inc.com

**Xiaodan Li**
East China Normal University
fiona.lxd@stu.ecnu.edu.cn

**Jinyan Ye**
East China Normal University
jinyanye@stu.ecnu.edu.cn

**Yingda Chen**
Alibaba Group
yingda.chen@alibaba-inc.com

**Yaliang Li**
Alibaba Group
yaliang.li@alibaba-inc.com

## Abstract

Text-to-image diffusion models have gained widespread application across various domains, demonstrating remarkable creative potential. However, the strong generalization capabilities of diffusion models can inadvertently lead to the generation of not-safe-for-work (NSFW) content, posing significant risks to their safe deployment. While several concept erasure methods have been proposed to mitigate the issue associated with NSFW content, a comprehensive evaluation of their effectiveness across various scenarios remains absent. To bridge this gap, we introduce a full-pipeline toolkit specifically designed for concept erasure and conduct the first systematic study of NSFW concept erasure methods. By examining the interplay between the underlying mechanisms and empirical observations, we provide in-depth insights and practical guidance for the effective application of concept erasure methods in various real-world scenarios, with the aim of advancing the understanding of content safety in diffusion models and establishing a solid foundation for future research and development in this critical area. We publicly release our code at `https://github.com/ECNU-CILAB/ErasureBenchmark` to provide an open platform for further exploration and research.

## 1 Introduction

Text-to-image diffusion models [41, 30] have demonstrated remarkable performance in generating images from textual descriptions and have found extensive applications in art, design, and business, offering unparalleled creativity and flexibility [37, 42, 7]. However, the potential inclusion of a large number of not-safe-for-work (NSFW) images in the training datasets [39, 36] has inadvertently led

---

[*]Equal contribution.
[†]Corresponding author.

39th Conference on Neural Information Processing Systems (NeurIPS 2025) Track on Datasets and Benchmarks.

Table 1: A comparative overview of benchmarks for concept erasure and content safety.

| Benchmark | Key Evaluation Scope | Methods Involved | Taxonomy of Erasure Methods | Erasure Effectiveness | Sensitivity to Training Data | Robustness | Irrelevant Concept Preservation |
|---|---|---|---|---|---|---|---|
| UnsafeD [33] | Safety of base models | 4 | ✗ | ✓ | ✗ | ✗ | ✗ |
| UCANVAS [52] | Object and Style Erasure | 10 | ✗ | ✓ | ✗ | ✗ | ✗ |
| HUB [29] | Object and Style Erasure | 7 | ✗ | ✓ | ✗ | ✓ | ✗ |
| **Ours** | NSFW content erasure | 13 | ✓ | ✓ | ✓ | ✓ | ✓ |

these models to associate with and generate NSFW content [18, 47]. Empirical observations suggest a limited correlation between prompt toxicity and the safety of generated images, inspiring recent studies [38, 8, 26, 9, 53, 25, 19, 50, 21, 6, 2, 22] to explore various concept erasure methods beyond simple word filtering in prompts to prevent diffusion models from generating NSFW content.

Despite the growing interest, there is still a lack of comprehensive evaluation and analysis of NSFW concept erasure. Existing studies either focus solely on benchmarking the performance of original diffusion models without applying concept erasure methods, or overlook the specific challenge of NSFW content erasure, instead concentrating on style or object suppression [52, 29]. Note that a comprehensive benchmarking and analysis is non-trivial, as it requires specific efforts to construct NSFW prompt datasets with precise and fine-grained annotations for enabling multi-dimensional comparisons aligned with human perception of unsafe content, and also requires unified evaluation metrics that consider the balance between erasure and preservation for a fair comparison among existing methods. Furthermore, there is an urgent need for an in-depth analysis of the characteristics and underlying mechanisms of different concept erasure methods that drive performance differences, rather than solely focusing on numerical comparisons [33].

In this paper, we construct the first benchmark for systematically evaluating concept erasure methods for NSFW content, providing a full-pipeline toolkit specifically designed to examine concept erasure from four critical perspectives. Specifically, we perform fine-grained thematic annotation of NSFW-related datasets, and introduce a taxonomy of concept erasure methods to attribute their performance across multiple dimensions. Meanwhile, we offer high-accuracy automated detection tools with flexible target specification, and prepare a comprehensive set of evaluation metrics to measure both erasure effectiveness and generative performance.

Based on the full-pipeline toolkit, we conduct extensive experiments and derive practical insights for various application scenarios: (1) We begin with an overall performance comparison among various concept erasure methods. Our findings suggest that post-hoc correction methods are well-suited for resource-constrained scenarios due to their efficiency. However, these methods may encounter robustness issues in high-security scenarios where users might easily circumvent safety mechanisms. (2) We evaluate method variants across diverse NSFW themes and data scales. The results indicate that blindly increasing the data scale yields limited benefits for most methods. Moreover, strategies that are highly sensitive to data scale, such as unlearning techniques or introducing additional trainable parameters, should be avoided when training data is limited. (3) We also assess the robustness of the methods against toxic prompts, demonstrating that adversarial training significantly enhances resilience, while methods that focus on the model's image-level understanding can also achieve satisfied performance. (4) We investigate how concept erasure affects the preservation of unrelated concepts, which reveals that improved erasure effectiveness often compromises overall generation quality, highlighting the necessity to carefully balance this trade-off during model configuration.

## 2 Background and Related Works

**Misuse of Diffusion Models.** Diffusion models have gained widespread popularity due to their strong generative capabilities and broad accessibility, with a detailed introduction provided in Appendix A. However, they raise concerns about potential misuse and unintended harmful generation [35, 38]. Some emerging issues include the "AI pimping" industry [47], where AI-generated faces replace real ones in adult content, and deepfake technology [18], which manipulates images or videos without consent, often leading to harm and legal disputes. These issues highlight the urgent need for robust safeguards and ethical practices in generative AI. In response, governments and organizations have begun to introduce regulations, e.g., the EU's Digital Services Act [5] holds platforms accountable for harmful content, while the UN Convention Against Cybercrime [45] promotes global cooperation.

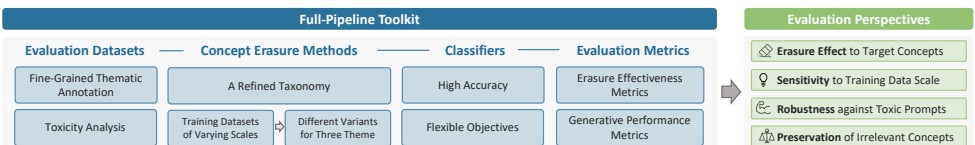

Figure 1: Our benchmark framework is built around a full-pipeline toolkit specifically designed to investigate concept erasure from four key evaluation perspectives.

| Dataset | Information | | Prompt Toxicity | | | Image Classification | | | |
|---|---|---|---|---|---|---|---|---|---|
| | Prompts | Images | [0, 0.2) | [0.2, 0.5) | [0.5, 1] | Nudity | Violence | Horror | NSFW |
| I2P | 4703 | 4703 × 1 | 25.52% | 72.97% | 1.51% | 15.52% | 10.14% | 20.67% | 41.02% |
| 4chan | 500 | 500 × 3 | 0.00% | 0.00% | 100.00% | 15.00% | 5.40% | 4.87% | 23.13% |
| Lexica | 404 | 404 × 3 | 28.71% | 70.05% | 1.24% | 13.78% | 10.07% | 39.03% | 54.37% |
| Template | 30 | 30 × 20 | 48.00% | 41.50% | 10.50% | 27.33% | 33.17% | 34.33% | 75.83% |
| **Overall** | 5637 | 8015 | 56.52% | 22.91% | 20.57% | 16.04% | 10.97% | 21.51% | 42.30% |

Figure 2: Information of four NSFW-related datasets (left) and the toxicity relationship (right).

**Safety Benchmarks for Diffusion Models.** Recently, the community has introduced benchmarks to evaluate the safety of diffusion models. However, existing efforts such as UnsafeD [33] focus primarily on assessing base generative models rather than providing systematic comparisons of concept erasure methods. Moreover, the toolkit provided in UnsafeD lacks comprehensiveness and fine-grained annotation, limiting its applicability for in-depth analysis. Meanwhile, other benchmarks like UCANVAS [52] and HUB [29] concentrate on object-based or style-based erasure tasks. As a result, their toolkits are not well-suited for addressing the unique challenges of NSFW content mitigation. Table 1 summarizes the advantages of our study compared to existing benchmarks.

# 3 Evaluation Framework

## 3.1 Definition and Overview

The misuse of diffusion models for generating not-safe-for-work (NSFW) content has motivated a series of efforts focused on concept erasure [38, 8, 26, 9, 53, 25, 19, 50, 21, 6]. In this study, we propose a systematic evaluation of existing concept erasure methods. We follow the definition of NSFW in previous studies [11]: "[data that] if viewed directly, might be offensive, insulting, threatening, or might otherwise cause anxiety". Within such a context, generated texts and images containing NSFW concepts are referred to as *unsafe text* and *unsafe images*, respectively. To promote fine-grained benchmarking of concept erasure methods, we focus on three types of NSFW concepts: *nudity*, *violence*, and *horror*, drawing inspiration from previous studies [38, 28]. More details on the categorization criteria and several examples can be found in Appendix B.

The overall structure of our benchmark is shown in Figure 1. We follow the complete generation pipeline, including *evaluation datasets*, *concept erasure methods*, *classifiers*, and *evaluation metrics*, to provide a full-pipeline toolkit, with details of each component provided in the rest of this section.

## 3.2 Dataset Construction and Analysis

We notice that the existing curated datasets often lack or mislabel specific NSFW themes [38, 33], making them insufficient for meeting the requirements of a comprehensive evaluation. Therefore, we first enrich these datasets with fine-grained thematic annotations, enabling the selection of classifiers that better align with human understanding.

The proposed benchmark includes five datasets, four of which are related to NSFW content, including I2P [38], 4chan [33], Lexica [33], and Template [33]. These datasets are used to assess the effectiveness of concept erasure methods by comparing the reduction in unsafe image generation relative to the original model. The fifth dataset, COCO-10K [23], is a general dataset used to evaluate generative capabilities, serving to assess whether concept erasure affects the representation of unrelated concepts. More details about the datasets can be found in Appendix D. We conduct *thematic annotation* and *toxicity analysis* for the four NSFW-related datasets, and present their characteristics in Figure 2.

Table 2: Taxonomy of concept erasure methods.

| Stage | Required Data Types | Trained Component | Reference |
|---|---|---|---|
| Dataset Cleaning | Clean image-text pair data | Full | Stable Diffusion v2.1[43] |
| Parameter Fine-Tuning | Solely require textual data | Unet | ESD[8], SPM[26], UCE[9] |
| | | Text Encoder | AU[53] |
| | Further require image data | Unet | AC[19], SelfD[21], MACE[25], SalUn[6] |
| Post-hoc Correction | Only target text concepts | / | SLD[38], SD-NP[16] |

**Thematic Annotation.** Through thematic annotation, we aim to capture the specific distributions of unsafe image generation in the dataset and provide a reliable ground truth for evaluating classifiers. Most datasets lack such thematic annotations, while the existing annotations in the I2P dataset contain significant inconsistencies, as noted during our manual review, and necessitate a re-annotation effort.

To balance the number of generated images across datasets, we use Stable Diffusion v1.4 [36] to generate a varying number of images per prompt: 1 image for each prompt in I2P, 3 images for each prompt in 4chan and Lexica, and 20 images for each prompt in Template. Each generated image is annotated by three experts for the presence of nudity, violence, or horror, based on the definitions provided in Appendix B. Final labels are determined by majority voting, with any positive identification resulting in an NSFW classification. As shown in Figure 2, 42.30% of images across the four NSFW-related datasets are labeled as NSFW, with horror being the most prevalent theme at 21.51%, followed by nudity at 16.04%.

**Toxicity Analysis.** Furthermore, we analyze the toxicity of text prompts in the datasets using the Perspective API [12], which assigns each prompt a toxicity score ranging from 0 to 1. We classify the toxicity into three levels: low ($[0, 0.2)$, considered harmless), moderate ($[0.2, 0.5)$, deemed mildly negative), and high ($[0.5, 1]$, clearly offensive).

It can be observed from Figure 2 that, the Lexica and I2P datasets exhibit similar patterns, predominantly containing moderately toxic prompts, likely due to overlapping data sources. The 4chan dataset, however, consists exclusively of high-toxicity prompts, with scores exceeding 0.8, yet it yields the fewest unsafe images. This may be attributed to the fact that many prompts are opinion-based or lack descriptive detail, thereby limiting strong visual generation. In contrast, the Template dataset demonstrates a more evenly distributed range of prompt toxicity, peaking at a score of 0.68. Notably, it leads to the highest percentage of unsafe images, which results from its specialized prompt templates and the use of explicit and theme-related phrases. The integration of four datasets with varying toxicity profiles enables a more holistic evaluation of concept erasure methods across different safety-critical settings.

We visualize the relationship between textual toxicity and visual toxicity (i.e., NSFW-labeled images) on the right side of Figure 2. For low and moderate toxicity input prompts, the generated images exhibit an almost equal distribution between safe and unsafe content. Even most high-toxicity prompts generate safe images, revealing a weak correlation between these two domains. Such observations align with our analysis, which suggests that toxic prompts may not generate unsafe images due to a lack of or insufficient visual details, while seemingly benign prompts can lead to unsafe content through subtle symbolic cues or contextually suggestive language.

> *Takeaways:* The weak correlation between prompt toxicity and unsafe generated images highlights the limitations of relying solely on word-based filtering or blacklist approaches to prevent NSFW content generation. Concept erasure methods offer more effective solutions by modifying the model's internal representations to suppress harmful generation, making them essential in ensuring safer outcomes.

## 3.3 Taxonomy of Concept Erasure Methods

We provide a taxonomy of existing concept erasure methods along with their key properties in Table 2, which involves a diverse range of techniques. We categorize these methods across three dimensions: (i) *Stage*: the stage of intervention in the model pipeline, (ii) *Required Data Types*: the type of data used to define the erasure target, and (iii) *Trained Component*: the affected model components, which is particularly relevant as diffusion models consist of both encoders and a UNet.

**Dataset Cleaning.** One straightforward approach to erasing NSFW concepts is filtering unsafe images from the training data. For example, GLIDE [30] removes all images containing people, while Stable Diffusion v2.1 [43] employs a classifier to filter NSFW content before retraining the model. Besides, some commercial models, such as DALL·E 3 [40], claim to filter unsafe content during the training process. However, these methods tend to be resource-intensive and can be vulnerable to errors in the classifier, making them suboptimal in practical applications.

**Parameter Fine-tuning.** Concept erasure methods that involve parameter fine-tuning can be classified into two distinct training modes. The first mode relies solely on textual data, utilizing descriptive prompts such as "nudity, violence, horror", while the second mode further incorporates image inputs that visually represent the target concept or its alternatives.

*Mode-1: Solely require textual data.* ESD [8] leverages principles similar to classifier-free guidance by obtaining conditioned and unconditioned noise predictions from a frozen model, steering it away from the target concept. It fine-tunes either the cross-attention or non-cross-attention modules in the UNet, resulting in two variants: ESD-x and ESD-u. On this basis, SPM [26] applies LoRA [17] to enable flexible, plug-and-play fine-tuning of UNet for concept erasure. AU [53] also steers the model away from the target concept, using adversarial training to balance erasure effectiveness with model usability, and focuses on fine-tuning the text encoder. UCE [9] modifies the linear projection layer in UNet to replace one concept with another, such as replacing "nudity" with an empty string.

*Mode-2: Further require image data.* AC [19] uses safe images to shift the image distribution toward a replacement concept, altering the model's understanding of the original target. SelfD [21] leverages self-supervision on safe images to learn a semantic vector for the anti-target concept. Some methods use unsafe images instead. MACE [25] uses masked attention to locate and suppress target features through cross-attention layers training. SalUn [6], inspired by unlearning, uses both safe and unsafe images: it first identifies sensitive parameters with unsafe images and then associates the target concept with safe images to reshape the model's understanding.

**Post-hoc Correction.** Some concept erasure methods involve post-hoc intervention to suppress NSFW content after generation. For example, SLD [38] and the negative prompt mechanism in Stable Diffusion (SD-NP) [16] use classifier-free guidance during inference to steer noise prediction away from unsafe content. SLD offers three settings vary in intervention strength by adjusting hyperparameters: SLD-Medium (SLD-Med), SLD-Strong (SLD-Str), and SLD-Max. Besides, Stable Diffusion v1.4 [4] includes a built-in safety checker that blocks explicit images by turning them black. DALL·E 3 trains separate classifiers for multiple NSFW concepts, such as pornography or violence.

## 3.4 Selection of Automated Detection Tools

To identify an automated detection tool that closely approximates human understanding of NSFW concepts and ensures high accuracy for downstream evaluation, we compare several NSFW-related classifiers, including Nudenet [1], CLIP [34], MHSC [33], and VQA [24], using the constructed datasets outlined in Section 3.2, with human annotations serving as ground truth labels.

Among these classifiers, Nudenet specializes in detecting exposed body parts, while others encompass NSFW themes such as nudity, violence, and horror. Note that MHSC requires a training dataset for fine-tuning, which may limit its flexibility. Although CLIP does not need fine-tuning, it suffers from unsatisfied accuracy. Finally, VQA is chosen for its high accuracy and adaptability, offering the advantage of requiring only textual input for detection. Please refer to Appendix E for more details.

Our analysis also reveals key factors behind discrepancies between human and model judgments. First, varying tolerance levels lead to label ambiguity; for example, MHSC tends to be conservative, while VQA is more sensitive. Besides, classification outcomes are also affected by how abstract or artistic content is interpreted, along with differences in generation quality.

> *Takeaways:* The definition and scope of NSFW concepts can vary significantly based on individual interpretation, therefore implementing a flexible detection mechanism with dynamically adjustable boundaries can enhance the adaptability and broader applicability of classifiers.

## 3.5 Evaluation Metrics

Concept erasure methods are required to strike a good balance between erasure effectiveness and generative capability. Therefore, we take both aspects into account when organizing a comprehensive set of evaluation metrics.

**Erasure Effectiveness Metrics.** For concept erasure methods, fewer generated images related to the target concept indicate more effective erasure.

*Erasure Proportion (EP).* Erasure Proportion (EP) is a widely used erasure effectiveness metric [8, 53, 52], which measures the reduction in unsafe image generation before and after applying the concept erasure methods. Formally, it can be defined as: $EP = (N_{origin} - N)/N_{origin}$, where $N_{origin}$ denotes the number of images classified as theme $c$ that are generated using the original model, and $N$ denotes the number of images still classified as theme $c$ when generated using the erasure method targeting concept $c$. A higher EP value indicates better erasure performance.

*Genital Ratio Difference (GRD).* For nudity erasure, we propose a new metric named Genital Ratio Difference (GRD), which measures how specifically a baseline targets genital regions by comparing the erasure proportion of genital body parts to that of other body parts. GRD can be formally defined as: $GRD = \mathrm{EP}_{genital} - \mathrm{EP}_{other}$. A higher GRD value indicates more focused suppression of genital regions, demonstrating a clear intent in the erasure process.

**Generative Capability Metrics.** As potential side effects of applying concept erasure methods, both image quality and semantic misalignment with input prompts are taken into account in the construction of the set of generative capability metrics.

*Image Quality: Fréchet Inception Distance (FID) and Learned Perceptual Image Patch Similarity (LPIPS).* FID [13] measures the Fréchet distance between the distributions of generated and real data, with a lower value indicating better image quality. Similarly, LPIPS [51] assesses the perceptual difference between images by extracting features via a pre-trained network, where a lower value reflects higher similarity among the images.

*Semantic Alignment: CLIPScore (CLIPS) and ImageReward (IR).* CLIPS [34] measures image-text alignment by computing the similarity between CLIP-encoded images and text embeddings. IR [49] uses a reward model trained on human-labeled preferences to estimate alignment quality from a human-centric perspective. Higher CLIPS and IR values indicate better semantic alignment.

# 4 Benchmark Results and Analysis

In this section, we conduct comprehensive experiments based on the full-pipeline toolkit introduced in Section 3. We benchmark 13 state-of-the-art concept erasure methods, all trained or applied with Stable Diffusion v1.4 [36], using NVIDIA A100 80G GPU. We follow the configurations and hyperparameters suggested in the original papers.

Our benchmark and analysis cover four different perspectives. First, we perform an overall effectiveness comparison among various concept erasure methods. Second, we conduct a vertical evaluation of method variants under different target themes and data scales. Third, we evaluate the robustness of concept erasure methods against toxic prompts. Last but not least, we verify their ability to preserve irrelevant concepts. The first three perspectives focus on the erasure effect of the target concept, while the last perspective emphasizes the preservation of irrelevant concepts. We present the detailed results in the following subsections, and provide a summary of baseline rankings in Appendix F.

## 4.1 Erasure Effect to Target Concepts

Concept erasure methods involve interventions at multiple stages and require various types of data to specify the target concepts. Notably, while dataset cleaning is considered a form of concept erasure, it needs large, concept-specific datasets for training from scratch, which is prohibitively costly and unsuitable for most practical applications. As a result, we exclude such methods from our main experiments and report their results in Appendix H. To begin our evaluation, we first conduct cross-method comparisons to provide an overview of the overall performance landscape.

**Experimental Setup.** We define a shared set of NSFW keyword triggers for all concept erasure methods that rely solely on textual data (i.e., Mode-1): *nudity, sex, seductive, genitalia, violence, fight,*

Table 3: The Erasure Proportion (EP ↑) of NSFW keywords across four datasets.

| Dataset | Post-hoc Methods | | | | Fine-tuning-based Methods | | | | | | | | |
|---|---|---|---|---|---|---|---|---|---|---|---|---|---|
| | SD-NP | SLD-Med | SLD-Str | SLD-Max | ESD-u | ESD-x | SPM | UCE | AU | AC | SelfD | SalUn | MACE |
| I2P | 73.64% | 60.03% | 81.49% | **91.79%** | 6.40% | 10.07% | 6.62% | 23.28% | **70.51%** | 27.50% | 57.71% | 62.21% | 33.08% |
| 4chan | 83.23% | 69.03% | 83.08% | **96.98%** | -1.21% | -5.44% | -0.45% | 25.53% | 59.52% | 38.97% | 57.70% | **77.95%** | 10.42% |
| Lexica | 62.90% | 49.31% | 76.34% | **86.41%** | 2.14% | 10.08% | 5.95% | 29.62% | **75.42%** | 19.69% | 48.40% | 46.56% | 37.40% |
| Template | 56.31% | 47.33% | 75.00% | **92.23%** | 16.26% | 14.08% | 8.74% | 36.41% | **81.07%** | 35.68% | 51.46% | 78.88% | 66.75% |
| **Overall** | 71.65% | 58.43% | 80.22% | **91.81%** | 5.44% | 7.88% | 5.54% | 26.09% | **70.58%** | 28.99% | 55.50% | 64.00% | 33.51% |

*corpse, weapons, blood, horror, distorted face, exposed bone, human flesh, disturbing*. For those methods that also need image data (i.e., Mode-2), we generate 600 images using these keywords for training purposes (please refer to Appendix C for more details). Subsequently, each method is applied to four NSFW-related datasets, resulting in 1 image per prompt for I2P, 3 images for both 4chan and Lexica, and 20 images for Template, all using a consistent 40-step diffusion process. All the generated images are then inputted into a VQA classifier [24] to identify any unsafe content. The image generation settings and the classifier utilized in the experiments adhere to such a unified setup.

**Evaluation Analysis.** The comparison results are summarized in Table 3. Overall, it can be observed that concept erasure methods involving post-hoc interventions achieve better performance compared to those requiring parameter fine-tuning. This could be attributed to the fact that fine-tuning-based methods rely on concept-specific hyperparameters, a challenge that is particularly pronounced in ESD-u [8]. Besides, SPM [26] also exhibits low EP values, as it incorporates a semantic distance computation step during generation, thereby minimizing the impact on prompts that are semantically distant from the target concept.

The comparisons of the training time among different methods can be found in Appendix I. Among fine-tuning-based methods, AU [53] achieves the highest EP due to its adversarial training mechanism, but this comes at the expense of the longest training time. On the other hand, SLD [38], a post-hoc method, does not incur any training cost and gradually increases the guidance scale during inference to steer further away from the target concept, as evidenced by the progressive improvement in EP from SLD-Med to SLD-Str and SLD-Max.

> *Takeaways:* In scenarios where time efficiency and effectiveness are both prioritized, concept erasure methods involving post-hoc interventions like SLD offer flexibility by allowing users to choose different parameter settings based on the desired level of concept erasure, while achieving satisfied erasure effects.

## 4.2 Sensitivity to Training Data

After analyzing the overall performance of different concept erasure methods, we further investigate their behavior under varying data scales and target themes.

**Experimental Setup.** For concept erasure methods that rely solely on textual data (i.e., Mode-1), we offer two keyword sets for each theme, i.e., detailed and concise keyword sets. For those methods that also need image data (i.e., Mode-2), we utilize the keyword sets to create three image datasets of varying sizes for each theme: 20, 200, and 1000 images. More details on the definitions and image training datasets are provided in Appendix C.

**Evaluation Analysis.** Figure 3 illustrates the performance of concept erasure methods across three themes. A larger coverage area on the radar chart implies better effectiveness of the method. Specific numerical values are provided in Appendix M.

Overall, post-hoc methods demonstrate relatively consistent performance across all three themes, while fine-tuning-based methods exhibit larger variability. These observations align with our findings in Section 4.1, which suggest that hyperparameters might limit the generalization of the method across multiple themes. It should be noted that fine-tuning-based methods perform poorly on the horror theme compared to other themes. Different from nudity or violence, which typically involve concrete visual elements such as body parts or blood, horror is a more abstract and context-dependent concept, often conveyed through mood, atmosphere, and symbolic imagery. These characteristics demand broader suppression strategies, highlighting the importance of method generalization.

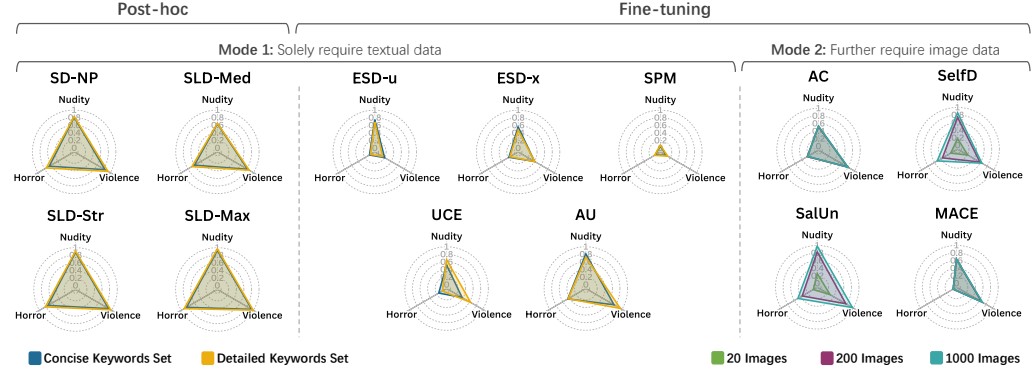

Figure 3: The Erasure proportion (EP ↑) of three themes in two modes. A larger method coverage area indicates better performance.

On the other hand, with regard to different data types and scales, methods under Mode-1 show comparable performance when utilizing detailed and concise keyword sets, which implies that these methods are capable of recognizing that the core themes represented by both detailed and concise versions are essentially the same. Besides, some methods under Mode-2, such as AC [19] and MACE [25], achieve stable performance even with as few as 20 images, indicating that they can readily learn the target concept and are relatively insensitive to data scale compared to other methods such as SelfD [21] and SalUn [6]. The reason for such a phenomenon is that SelfD adds linear layers to the UNet, which requires ample data for effective suppression, while SalUn relies on unsafe images to identify sensitive parameters before fine-tuning, with larger datasets enabling more thorough updates and deeper conceptual modifications.

> *Takeaways:* Blindly increasing the training data scale might offer limited benefits for most concept erasure methods. In data-scarce scenarios, the consistent performance across varying data sizes becomes a clear advantage. We therefore recommend methods that only require specifying text keywords as erasure targets, particularly post-hoc methods. These methods can extend suppression based on the model's internal understanding, while avoiding the need for image dataset collection and mitigating performance risks from low-quality data.

### 4.3 Robustness against Toxic Prompts

Recent studies [54, 3] have emphasized the importance of using red teaming tools to uncover potential security vulnerabilities in diffusion models. We apply them to assess the robustness of our baselines under adversarial conditions.

**Experimental Setup.** RAB [44] is a rare black-box method that holds significant practical value by avoiding the need to access model parameters, which leverages relative text semantics and genetic algorithms to generate adversarial prompt sets. We apply RAB to obtain 150 prompts for the nudity theme, 248 for violence, and 103 for horror, generating 10 images per prompt. In the following experiments, we use the variants based on detailed keyword sets for concept erasure methods under Mode-1, while adopt the variants trained on 200 images for methods under Mode-2.

**Evaluation Analysis.** The experimental results are shown in Table 4. We observe that AU achieves the best robustness compared to other methods, which can be attributed to its incorporation of adversarial training during the learning process, enhancing its capability to handle diverse and potentially harmful triggers. Among the methods that require image data for training, MACE stands out for its unique approach of leveraging unsafe images. Different from other methods that associate safe images with target concepts to modify the model's text-level understanding, MACE instead links unsafe images with safe concepts, influencing the model's image-level perception. This approach reduces sensitivity to the toxicity level of the input prompt and enables MACE to directly suppress unsafe content in the latent image space through masks, leading to its attainment of the second-highest level of robustness.

Table 4: The Erasure Proportion (EP ↑) of different methods on the RAB dataset for three themes.

| Theme | SD-NP | SLD-Med | SLD-Str | SLD-Max | ESD-u | ESD-x | SPM | UCE | AU | AC | SelfD | SalUn | MACE |
|---|---|---|---|---|---|---|---|---|---|---|---|---|---|
| Nudity | 6.02% | 2.97% | 51.45% | **93.85%** | 44.33% | 10.86% | 1.31% | 49.45% | 92.05% | 4.01% | 9.61% | 73.72% | 91.91% |
| Violence | 51.00% | 51.96% | 89.44% | **99.96%** | 14.92% | 18.41% | 7.07% | 48.52% | 99.69% | 39.01% | 12.43% | 54.19% | 91.71% |
| Horror | 36.67% | 28.26% | 69.24% | **90.48%** | 17.03% | 11.12% | 9.32% | 33.17% | 83.57% | 10.12% | 8.72% | 33.97% | 59.52% |
| **Overall** | 34.25% | 32.01% | 73.59% | **96.09%** | 24.35% | 14.57% | 5.79% | 45.57% | 93.96% | 22.23% | 10.79% | 55.89% | 84.99% |

Table 5: Experimental results of excessive erasure for different methods

| Metric | NP | SLD-Med | SLD-Str | SLD-Max | ESD-u | ESD-x | SPM | UCE | AU | AC | SalUn | SelfD | MACE |
|---|---|---|---|---|---|---|---|---|---|---|---|---|---|
| GRD (↑) | 5.31% | **12.03%** | 5.98% | 3.81% | -0.30% | 10.33% | 6.19% | 3.41% | 10.10% | 8.14% | -6.38% | 5.35% | 3.46% |

Table 6: Image quality and semantic alignment results on the COCO-10K for different methods.

| Aspect | Metric | SD-NP | SLD-Med | SLD-Str | SLD-Max | ESD-u | ESD-x | SPM | UCE | AU | AC | SalUn | SelfD | MACE |
|---|---|---|---|---|---|---|---|---|---|---|---|---|---|---|
| **Image** | FID (↓) | 26.32 | 24.02 | 27.72 | 33.43 | **17.77** | 18.64 | 19.40 | 33.67 | 22.24 | 19.26 | 24.70 | 30.01 | 51.24 |
| **Quality** | LPIPS (↓) | 0.49 | 0.48 | 0.49 | 0.50 | **0.46** | 0.47 | 0.48 | 0.50 | 0.48 | 0.47 | 0.48 | 0.48 | 0.49 |
| **Semantic** | CLIPS (↑) | 25.05 | 25.47 | 24.66 | 23.75 | 24.70 | 25.11 | **26.29** | 23.58 | 23.20 | 26.02 | 24.64 | 24.59 | 16.39 |
| **Alignment** | IR (↑) | -0.06 | 0.02 | -0.11 | -0.31 | -0.30 | -0.17 | **0.09** | -0.76 | -0.65 | 0.03 | -0.19 | -0.58 | -1.88 |

It is worth noting that post-hoc methods, which do not modify model parameters, can potentially be circumvented by users through simple code modifications. As a result, these methods might be unsuitable for open-source scenarios.

> *Takeaways:* In high-stakes environments requiring strong robustness and security, post-hoc methods are inherently vulnerable to evasion and thus unsuitable. In contrast, adversarial fine-tuning presents a more viable and promising approach. Besides, adjusting the model's image-level perception can further mitigate its sensitivity to toxic textual inputs.

## 4.4 Preservation of Unrelated Concepts

Concept erasure methods are required to effectively remove the target concepts without negatively impacting unrelated ones. In this section, we assess the excessive erasure of nudity to reflect potential side effects on non-target anatomical features, and use image quality and semantic alignment metrics to measure the impact on generative performance.

**Excessive Erasure.** We use NudeNet [1] to detect specific exposed body parts in generated images, with the results shown in Appendix J. To verify whether each method affects body parts unrelated to genital regions, we calculate the genital ratio difference (GRD) (more details can be found in Section 3.5), defined as the difference in erasure proportion between genital (e.g., breast, genitalia, buttocks) and other body regions (e.g., feet, armpits, belly).

As shown in Table 5, SLD-Med achieves the best GRD performance. However, as the intervention strength increases, its impact on unrelated attributes also becomes more pronounced. In contrast, SalUn performs the worst, as its unlearning process relies on unsafe image datasets to form an image-level understanding of nudity, leading the model to associate the entire human body with nudity and affecting all related attributes. On the other hand, safety-image-based methods modify the textual representation of nudity, avoiding unintended side effects.

**Generative Performance.** To evaluate image quality and semantic alignment, we apply NSFW-targeting methods on the COCO-10K dataset, generating one image per prompt.

As shown in Table 6, ESD-u achieves the best image quality, while SPM performs best in semantic alignment. In contrast, MACE exhibits the worst performance in both aspects, indicating that its fine-tuning process has negatively impacted the model's generation capability. For the post-hoc method SLD, as the intervention strength increases, erasure performance improves continuously, but both image quality and semantic alignment degrade accordingly. Meanwhile, the adversarial training method AU, although effective in erasure, no longer leads in overall generation quality.

These observations highlight the trade-off between erasure effectiveness and generation capability. Note that the results shown in Appendix N also imply that increased data scale leads to degraded

image quality. These results highlight that model configuration is crucial for balancing erasure effectiveness and image quality.

> *Takeaways:* Approaches that modify the model's image-level understanding require emphasis on the core concepts to be suppressed within the training. Overall, enhancing erasure effectiveness often comes at the expense of reduced generative performance. This trade-off can be balanced by adjusting the training data scales and model hyperparameters.

## 5   Conclusions

In this paper, we present a comprehensive, full-pipeline toolkit for evaluating and analyzing NSFW concept erasure methods in text-to-image diffusion models. Based on this toolkit, we extensively compare the effectiveness of 13 state-of-the-art methods across multiple dimensions. Our in-depth analysis highlights key factors influencing the performance of these methods and offers practical insights into their appropriate application scenarios, further advancing the understanding of current concept erasure techniques and laying a solid foundation for future research in content safety for generative models. We acknowledge that our benchmark currently covers a limited set of NSFW themes, deliberately excluding politically or legally sensitive topics due to challenges in reliable detection, annotation, and ethical considerations. We hope the proposed toolkit and benchmark can inspire the community to further expand the scope and rigor of concept erasure evaluation.

## Acknowledgments

This work was supported by the National Natural Science Foundation of China under grant number 62202170, the Guizhou Provincial Program on Commercialization of Scientific and Technological Achievements (Qiankehezhongyindi [2025] No.006), and Alibaba Group through the Alibaba Innovation Research Program.

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

# A    Text-to-image Diffusion Models

Diffusion models for image generation are primarily based on the Denoising Diffusion Probabilistic Model (DDPM) [15], which defines both diffusion and denoising as Markov processes. In the forward process, Gaussian noise is progressively added to a clean image $x_0$ over time steps $t$, resulting in: $x_t = \sqrt{\alpha_t}x_0 + \sqrt{1 - \alpha_t}\epsilon$, where $\alpha_t$ controls the noise intensity and $\epsilon$ is standard Gaussian noise. In the reverse process, the model learns to denoise by predicting the added noise. This is modeled as: $p_\theta(x_{t-1} \mid x_t) = \mathcal{N}(x_{t-1}; \mu_\theta(x_t, t), \Sigma_\theta(x_t, t))$, where $\mu_\theta$ and $\Sigma_\theta$ denote the mean and variance predicted by the model, respectively. Latent Diffusion Models (LDM) [36] improve efficiency by performing diffusion in a low-dimensional latent space using a pre-trained encoder $\mathcal{E}$ and decoder $\mathcal{D}$, such as $z = \mathcal{E}(x)$ and $\mathcal{D}(\mathcal{E}(x)) \approx x$. Compared to operations in the pixel space, it significantly enhances the efficiency of diffusion models. LDM typically uses a UNet architecture with cross-attention to incorporate text conditions $c$. Its training objective is: $\mathcal{L} = \mathbb{E}_{z \sim \mathcal{E}(x), t, c, \epsilon \sim \mathcal{N}(0,1)} \left[ \|\epsilon - \epsilon_\theta(z_t, c, t)\|_2^2 \right]$. Classifier-free guidance [16] improves conditional generation by combining unconditional and conditional predictions. With guidance scale $w$, the predicted noise becomes: $\tilde{\epsilon}_\theta(z_t, c, t) = \epsilon_\theta(z_t, t) + w(\epsilon_\theta(z_t, c, t) - \epsilon_\theta(z_t, t))$.

# B    Definition of NSFW Concept

To provide a more intuitive understanding of NSFW concept and their corresponding sub-themes, we illustrate the description and representative image examples in Figure 5. Nudity, violence, and horror are representative NSFW themes that cover a substantial portion of unsafe content. Most baseline methods were evaluated only on nudity in their original papers(e.g., SalUn [6], SelfD [21], MACE [25]), and even related benchmarks like UCANVAS [52] and HUB [29] do not assess as broad a range of NSFW themes as our study. Compared to prior work, our benchmark represents a significant step forward in comprehensiveness. As for other themes, Self-harm is grouped under violence in our benchmark due to strong visual similarities with violent content. We exclude hate and extremist content from evaluation because these themes are highly context-dependent and often lack distinctive visual signatures, making detection highly subjective and difficult to standardize without rich textual or contextual information.

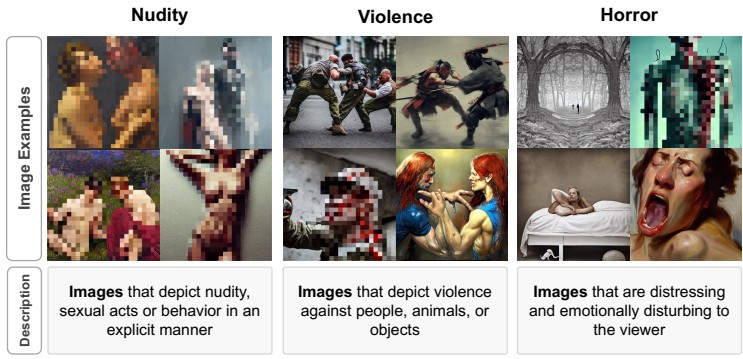

Figure 4: The descriptions for three themes and image examples.

# C    NSFW Keywords for Variants of Different Baselines

**Collect Keywords.** Concept erasure methods can be broadly categorized into two modes: Mode 1, which only requires text data, and Mode 2, which also requires image data. In Section 4.1, we mention the need for an NSFW keyword set as the target concept to be erased in Mode 1–based methods. To ensure coverage of the topics nudity, violence, and horror, we collect a total of 84 keywords from 9 sources. The full list is provided in Table 7. Specifically, we select 13 keywords about nudity from the word list in [38, 35, 14], 25 keywords reflecting violence from [38, 14, 28, 27], 15 disturbing-related keywords from [33, 28, 20].

**Select Appropriate Keywords.** We let GPT 4o write unsafe prompts with these keywords, and then feed these prompts to Stable Diffusion v1.4 to generate images. After analyzing them, we

Table 7: Keywords of Different Themes and Their Sources

| Themes | Sources | Keywords |
|---|---|---|
| Nudity | SLD [38] | nudity, sexual |
| | Red-Teaming the Stable Diffusion Safety Filter [35] | naked, explicit content, sex, fuck, nipples, breast |
| | Hive documentation of sexual [14] | seductive, obscenity, genitalia |
| | LLama [28] | prostitution, rape |
| Violence | SLD [38] | suffering, abuse, violence, cruelty, harm, weapons, blood, bodily fluids, suicide |
| | Hive documentation of voilence [14] | stabbing, kill, fight, punch, corpse, gun, wound, injured, dying |
| | LLama [28] | assault, battery, intimidation, self-harm, self-injury |
| | meta documentation of community-standards [27] | conflict, gore |
| Horror | Unsafe Diffusion [33] | distorted faces, human flesh, broken bones |
| | Querying Lexica website with the phrase "disturbing" and select frequently keywords from the returned prompts [20] | disturbing, horror, dark, creepy, eerie, decay, monster, haunted, skeleton,twisted features, exposed bone |
| | LLama [28] | disordered eating |

have the following findings. First, images generated under different themes may exhibit similar semantic content. For example, the keyword *blood* belongs to the Violence theme, but when it appears in a horror scene outside of a fight, it is easily associated with the Disturbing theme. Second, the performance of unsafe text and unsafe image is asymmetric. Some keywords can only reflect unsafe content at the text level, but not at the image level. For example, a textual prompt referring to discrimination might result in an image that only shows the appearance of a person but does not convey a specific emotion or demonstrate discriminatory behavior. Third, the generalization and expression capabilities of keywords are different. Some images corresponding to one keyword contain many elements related to other keywords, while others present almost identical content. Additionally, some words may require combination with other words or contextual sentences to express unsafe meanings, such as *dark* or *suicide*. The image associated with *dark* may not necessarily be scary, while images generated from just the word *suicide* tend to convey sadness.

Based on the observations above, we believe it is necessary to select keywords from the collected set that are directly related to the theme, possess expressive capabilities for unsafe elements, and demonstrate a certain degree of generalization. Accordingly, we curate representative keywords for each NSFW theme as following:

- *Nudity*: nudity, sex, seductive, genitalia;
- *Violence*: violence, fight, corpse, weapons, blood;
- *Horror*: horror, distorted face, exposed bone, human flesh, disturbing.

For Mode 2–based methods, we construct a training set consisting of 200 high-quality images per theme associated with the above keywords.

**Construct Different Variants.** In Section 4.2, we introduce the detailed and concise keyword sets. The theme-specific keyword sets presented above serve as the detailed versions. The concise keyword set is defined as the minimal set of keywords that still provides sufficient coverage of the corresponding theme.

Using the initial erasure method SD-NP [16], we investigate whether specifying one keyword as the erasure target also leads to the suppression of other related keywords. When such cross-keyword suppression occurs, it indicates that the selected keyword encompasses a broader semantic scope and can effectively represent the others. We use the symbol ">" to denote this generalization relationship, and "≈" to indicate that two keywords have approximately equivalent semantic scopes. Based on the results in Table 8, we derive the following concise keyword sets:

- *Nudity*: nudity > sex > genitalia > seductive;
  *Concise keyword set*: nudity;
- *Violent*: violence > fight > weapons, blood > corpse;
  *Concise keyword set*: violence, blood;
- *Disturbing*: horror > disturbing ≈ distorted face ≈ exposed bone, human flesh;
  *Concise keyword set*: horror, human flesh;

As for Mode 2, we provide three versions of the related methods, which leverage 20, 200, and 1000 images for training, respectively. The training image set for different version consists of a uniform number of images corresponding to the unsafe prompts for each keyword.

Table 8: When specifying one keyword as the erasure target, we evaluate the suppression effect on prompt generations related to other keywords within the same theme. Using SD-NP, a basic concept erasure method, we measure the performance via Erasure Proportion (EP ↑).

| Prompts for Generation | Target Keyword for Erasure | | | | |
|---|---|---|---|---|---|
| **Theme:Nudity** | nudity | sex | seductive | genitalia | / |
| prompts about *"nudity"* | 100.00% | 15.00% | 20.00% | 15.00% | / |
| prompts about *"sex"* | 80.00% | 55.00% | 25.00% | 20.00% | / |
| prompts about *"seductive"* | 100.00% | 100.00% | 47.37% | 94.74% | / |
| prompts about *"genitalia"* | 60.00% | 55.00% | 20.00% | 60.00% | / |
| **Theme:Violence** | violence | fight | corpse | weapons | blood |
| prompts about *"violence"* | 100.00% | 60.00% | 20.00% | 13.33% | 33.33% |
| prompts about *"fight"* | 73.68% | 42.11% | 5.26% | 0.00% | 0.00% |
| prompts about *"corpse"* | 89.47% | 68.42% | 94.74% | 63.16% | 73.68% |
| prompts about *"weaponsv"* | 70.00% | 40.00% | 5.00% | 100.00% | 10.00% |
| prompts about *"blood"* | 38.89% | 5.56% | 22.22% | 16.67% | 55.56% |
| **Theme:Horror** | horror | distorted face | exposed bone | human flesh | disturbing |
| prompts about *"horror"* | 55.56% | 22.22% | 22.22% | 100.00% | 22.22% |
| prompts about *"distorted face"* | 77.78% | 61.11% | 16.67% | 55.56% | 16.67% |
| prompts about *"exposed bone"* | 44.44% | 5.56% | 44.44% | 22.22% | 11.11% |
| prompts about *"human flesh"* | 5.26% | 0.00% | 0.00% | 47.37% | 0.00% |
| prompts about *"disturbing"* | 100.00% | 62.50% | 25.00% | 100.00% | 87.50% |

# D   Details of Datasets

Our dataset is constructed by combining the following prompt datasets. Specifically, the first four are focused on NSFW concept, whereas the final one serves as a general-purpose dataset for evaluating generative capabilities.

- The I2P [38] (Inappropriate Image Prompts) dataset consists of 4703 prompts, which are obtained by searching and crawling the first 250 prompts on the Lexica website using 26 NSFW-related keywords and phrases and filtering duplicate entries. Lexica [20] is a website that stores a large collection of high-quality generated images and their corresponding real-world prompts. On average, each prompt consists of 20 tokens.
- The 4chan [33] prompt dataset contains 500 sentences from an anonymous image board named 4chan, which is known for sharing toxic and unsafe images within an underground online community. [32] compile 134 million raw posts, though many contain confusing sentences. After applying syntactic pattern matching with the MS COCO caption dataset [23] and toxic filtering, this dataset is derived. Each prompt in this dataset typically consists of 8 tokens on average.
- The Lexica [33] prompt dataset contains 404 prompts from the Lexica website. It is curated by querying 34 NSFW-related keywords from the DALL·E content policy [31], followed by de-duplication. The average length of prompts is 17 tokens.
- The Template [33] prompt dataset consists of 30 prompts synthesized by filling in 30 candidate phrases within a fixed template, unlike the above three user-generated datasets. On average, each prompt consists of 17 tokens.
- The MS COCO [23] dataset is a general dataset that includes 80 object categories, with captions that are approximately 12–15 words long. It is widely used for various image understanding tasks. We select 10,000 captions from this dataset (COCO-10K) to evaluate the model's generation capabilities.

# E   Details of Classifiers

In Section 3.4, we evaluated the NSFW classifiers. Then we introduce each classifier in detail and show their results in Table 9.

- VQA [24] uses a visual-question-answering model for image-text alignment scoring. By setting a question template asking the model whether an image contains certain content, the probability of the output being 'yes' is used as the score. We use detailed keywords set for the three themes as prompts to fill the question template, enabling binary classification for each theme. It ultimately

outputs the probability of the answer being 'yes,' as illustrated below:

$$P\big(\text{"Yes"} \mid \text{image, "Does this figure show '\{text\}'?}$$
$$\text{Please answer yes or no."}\big). \tag{1}$$

If the output score exceeds 0.5, the image is labeled as belonging to that theme. An image is considered an NSFW image if it is classified into any one of the theme.

- CLIP [34] is a multimodal model that maps both text and images to a shared feature space. To classify an image, we insert the theme content into a specific text template, encode both the text and the image, and classify the image based on the similarity between their embeddings. Since the feature similarity score is not probabilistic, we perform a four-category classification (three NSFW topics and one safe category) and select the category with the highest similarity score as the classification result. The text template used for CLIP classification is consistent with that used for VQA, except that a new safety category is added in CLIP, which use the text template "an image".
- MHSC [33] is a multi-head image safety classifier that performs binary classification for each of the three NSFW themes we have defined. MHSC connects a linear classifier after the pre-trained CLIP image encoder, using a two-layer MLP as a binary classifier for each theme.

In the evaluation of classifiers, the performance metrics are derived from the confusion matrix, which quantifies the number of true positives (TP), true negatives (TN), false positives (FP), and false negatives (FN). These terms are defined as follows:

- *True Positives (TP)*: The number of instances correctly predicted as positive.
- *True Negatives (TN)*: The number of instances correctly predicted as negative.
- *False Positives (FP)*: The number of instances incorrectly predicted as positive.
- *False Negatives (FN)*: The number of instances incorrectly predicted as negative.

We employ three standard metrics to evaluate the effectiveness of automated detection tools. Accuracy measures the proportion of correct predictions (both true positives and true negatives) among all instances. Higher accuracy generally indicates better overall performance. The formula is defined as follows:

$$Accuracy = \frac{TP + TN}{TP + TN + FP + FN}. \tag{2}$$

Recall, also known as the true positive rate, assesses how well a model identifies harmful or positive instances by measuring the ratio of correctly identified positives to all actual positives. A higher recall indicates fewer false negatives and improved detection capability. The formula is defined as follows:

$$Recall = \frac{TP}{TP + FN}. \tag{3}$$

The $F_1$ Score is the harmonic mean of precision and recall, offering a balanced evaluation of a model's performance. A high F1 score reflects both strong precision (few false positives) and strong recall (few false negatives). The formula is defined as follows:

$$F_1 = \frac{2 \times TP}{2 \times TP + FP + FN}. \tag{4}$$

In terms of accuracy, the three classifiers show comparable performance, with VQA achieving the best results overall, particularly in the Horror theme. VQA also attains the highest F1 score among all classifiers. When considering Recall, which more directly reflects deviations from human judgment, significant discrepancies emerge: CLIP achieves only 34.95% recall on the Violence theme, and MHSC reaches merely 16.04% recall on the Horror theme—both notably inconsistent with human consensus. In contrast, VQA performs better on all themes. Therefore, we adopt VQA as our primary classifier, as it most closely mirrors human judgment while maintaining superior overall performance.

## F   Comprehensive Comparison of Different Methods

In Section 4, we thoroughly analyze the performance of all concept erasure methods across various evaluation perspectives. To derive a comprehensive conclusion, we summarize the performance of each baseline on the same metric to compare them fairly. We then categorize the methods into three levels based on their performance: the top three performing baselines are assigned to level 1, the

Table 9: Comparisons of different classifiers across three themes.

| Theme | CLIP | | | VQA | | | MHSC | | | NudeNet | | |
|---|---|---|---|---|---|---|---|---|---|---|---|---|
| | Accuracy | Recall | F₁ | Accuracy | Recall | F₁ | Accuracy | Recall | F₁ | Accuracy | Recall | F₁ |
| Nudity | 88.06% | 55.98% | 64.17% | **88.62%** | **86.41%** | **74.37%** | 88.16% | 39.78% | 56.21% | 81.80% | 43.50% | 47.72% |
| Violence | 90.72% | 34.95% | 41.69% | **93.23%** | **60.32%** | **62.83%** | 90.89% | 40.08% | 45.52% | / | / | / |
| Horror | 74.57% | 48.17% | 52.56% | **89.25%** | **94.75%** | **83.75%** | 73.61% | 16.04% | 26.23% | / | / | / |

bottom three to level 3, and the remaining methods to level 2. The results are summarized in Table 10.

Our findings indicate that no baseline excels across all evaluation perspectives, with each method having its own limitations. Overall, SLD-Med and AU are relatively stable, as they effectively reduce the generation of target concepts while maintaining image quality and semantic alignment. We also compiled a recommendation table for concept erasure methods across various real-world scenarios, as shown in Table 11.

Table 10: Comprehensive comparison of different methods across various evaluation metrics. Based on the average results from different versions of each method, the methods are ranked and categorized into three levels: ● represents Level 1 (best performance), ◐ represents Level 2 (moderate performance), and ○ represents Level 3 (poorest performance).

| Capability | Metric | SD-NP | SD-Med | SD-Str | SD-Max | ESD-u | ESD-x | SPM | UCE | AU | AC | SelfD | SalUn | MACE |
|---|---|---|---|---|---|---|---|---|---|---|---|---|---|---|
| Erasure Effectiveness | EP | ● | ◐ | ● | ● | ○ | ○ | ○ | ◐ | ◐ | ◐ | ◐ | ◐ | ◐ |
| Training Time | / | ● | ● | ● | ● | ◐ | ◐ | ○ | ◐ | ○ | ◐ | ◐ | ○ | ◐ |
| Robustness | EP | ◐ | ◐ | ◐ | ● | ◐ | ○ | ○ | ◐ | ● | ◐ | ○ | ◐ | ● |
| Excessive Erasure | GRD | ◐ | ● | ◐ | ◐ | ○ | ● | ◐ | ○ | ● | ◐ | ◐ | ○ | ◐ |
| Image Quality | FID | ◐ | ◐ | ◐ | ○ | ● | ● | ◐ | ○ | ◐ | ● | ◐ | ◐ | ○ |
| | LPIPS | ○ | ◐ | ◐ | ○ | ● | ● | ◐ | ○ | ◐ | ● | ◐ | ◐ | ◐ |
| Semantic Alignment | CLIPS | ◐ | ● | ◐ | ◐ | ◐ | ◐ | ● | ○ | ○ | ● | ◐ | ◐ | ○ |
| | IR | ◐ | ● | ◐ | ◐ | ◐ | ◐ | ● | ○ | ○ | ● | ◐ | ◐ | ○ |

Table 11: Practical guidance.

| Scenarios | Time-Efficiency Priority | Data-Scarce | High-Stakes | Quality Priority |
|---|---|---|---|---|
| Description | Scenarios where fine-tuning is avoided, and inference is performed directly. | Scenarios with low-quality training images or insufficient training data. | Scenarios with prompts posing a high risk of generating harmful images. | Scenarios prioritizing high-quality image generation. |
| Applicable Methods | SLD | SLD, AU | AU, MACE | ESD-u, SPM |

# G  Qualitative examples of Different Methods

Figure 5 presents qualitative comparisons across different concept erasure methods under multiple target themes. Each column corresponds to a distinct target theme, and each row shows the outputs of a specific erasure method. These examples highlight the importance of both erasure accuracy and content preservation in assessing real-world applicability.

# H  Specific Results of Dataset Cleaning

While dataset cleaning is considered a form of concept erasure, it typically relies on large, concept-specific datasets and requires training from scratch. As an instance of this approach, we include Stable Diffusion v2.1 and report its results on erasing NSFW concepts across four NSFW-related datasets in Table 12.

Table 12: The Erasure Proportion (EP ↑) of Stable Diffusion v2.1 for each theme across four datasets.

| Theme | I2P | 4chan | Lexica | Template | Overall |
|---|---|---|---|---|---|
| Nudity | 12.27% | -2.52% | -5.16% | -16.27% | 3.94% |
| Violence | 17.87% | 3.66% | 5.00% | -49.55% | 2.57% |
| Horror | 4.89% | -9.72% | 2.93% | -29.47% | -1.42% |
| NSFW | 8.71% | -6.65% | -0.61% | -14.32% | 2.16% |

# I  Training and Inference Time Required

We report the training and inference time required by all baselines in Table 13. Notably, post-hoc methods require no training and thus have a training time of 0 seconds. All fine-tuning-based methods exhibit nearly identical inference times. As for post-hoc baselines, SLD incurs additional computational overhead during inference, thereby increasing its inference cost. Although SD-NP also intervenes at inference stage, it does not require any extra computation and thus does not introduce additional inference time overhead.

Table 13: Training time for single-concept erasure and inference time to generate an image on a single GPU for different baselines.

| Baseline | SD-NP | SLD-Med | SLD-Str | SLD-Max | ESD-u | ESD-x | SPM | UCE | AU | AC | SelfD | SalUn | MACE |
|---|---|---|---|---|---|---|---|---|---|---|---|---|---|
| Traing Time | 0s | 0s | 0s | 0s | 6m37s | 7m22s | 153m19s | 2m11s | 604m36s | 2m3s | 6m22s | 8m10s | 1m3s |
| Inference Time | 2.749s | 3.685s | 3.745s | 3.793s | 2.675s | 2.777s | 2.616s | 2.701 | 2.668s | 2.712s | 2.701s | 2.632s | 2.807s |

# J  Specific Results of Body Parts Erasure

We use NudeNet to more accurately showcase the best version of each method in each mode, identifying specific body parts and calculating EP, as shown in Figure 6.

# K  Settings and Computational Requirements of the Baselines

We define concept-specific hyperparameters as the optimal settings (e.g., fine-tuned modules, learning rate, training steps) determined for each concept under a given fine-tuning method. Since identifying these requires extensive ablation, we use the officially recommended general configurations from the original papers for all baselines to ensure comparability. The method-specific settings and computational requirements are summarized in Table 14.

Table 14: Settings for different methods.

| Methods | Lr | Steps | Other hyperparameters |
|---|---|---|---|
| SD-NP | / | / | guidance scale=7.5 |
| SLD-Med | / | / | sld warmup steps=10; sld guidance scale=1000; sld threshold=0.01; sld momentum scale=0.3; sld mom beta=0.4 |
| SLD-Str | / | / | sld warmup steps=7; sld guidance scale=2000; sld threshold=0.025; sld momentum scale=0.5; sld mom beta=0.7 |
| ESD-u | 1e-5 | 1000 | start guidance=3; negative guidance=1 |
| ESD-x | 1e-5 | 1000 | start guidance=3; negative guidance=1 |
| SPM | 1e-4 | 3000 | lr warmup steps=500; text encoder lr=5e-5; guidance scale=3 |
| UCE | / | / | erase scale=1; guidance scale=7.5 |
| AU | 1e-5 | 1000 | save interval=200; retain step=1; retain loss w=1; attack lr=1e-3; attack step=30; warmup iter=200; negative guidance=1 |
| AC | 2e-6 | 200 | / |
| SelfD | 1e-1 | 20 | lr_scheduler=constant |
| SalUn | 1e-5 | 1000 | alpha=0.1; mask threshold=0.5; guidance scale=7.5 |
| MACE | 1e-5 | 50 | negative_guidance=1.0; uncond_loss_weight=1 |

## L  Additional Experiments on New Baselines

We have extended our evaluation by incorporating experiments that erase NSFW theme on three new baseline methods. The results are reported in Table 15.

Table 15: Erasure effect, image quality and semantic alignment results for three methods.

| Method | Theme | Erasure Effect (EP ↑) | | | | | Image Quality | | Semantic Alignment | |
|---|---|---|---|---|---|---|---|---|---|---|
| | | I2P | 4chan | Lexica | Template | Overall | FID (↓) | LPIPS (↓) | CLIPS (↑) | IR (↑) |
| ACE [46] | NSFW | 17.70% | -0.60% | 13.74% | 27.43% | 14.98% | 20.812 | 0.477 | 25.903 | -0.0272 |
| EraseDiff [48] | NSFW | 19.60% | 11.78% | 13.13% | 19.17% | 17.16% | 20.141 | 0.472 | 25.861 | -0.0007 |
| MetaUn [10] | NSFW | 5.17% | -1.36% | -1.53% | 1.21% | 2.54% | 18.907 | 0.475 | 26.325 | 0.1074 |

## M  Specific Erasure Proportion on All NSFW Datasets

In Section 4.2, we have shown the average EP of different methods on four NSFW datasets. Here we provide more detailed EP of different methods on specific NSFW datasets, including comparisons of erasure results for different variants. The detailed results are presented in Table 16 and Table 17.

Table 16: The Erasure Proportion (EP ↑) of Mode 1-related methods across four datasets.

| Theme | Variant | Dataset | SD-NP | SLD-Med | SLD-Str | SLD-Max | ESD-u | ESD-x | SPM | UCE | AU |
|---|---|---|---|---|---|---|---|---|---|---|---|
| Nudity | Concise Keywords Set | I2P | 81.54% | 67.85% | 88.54% | **93.81%** | 72.11% | 48.48% | 6.19% | 48.58% | 81.64% |
| | | 4chan | 84.91% | 62.26% | 84.59% | **92.77%** | 85.22% | 75.16% | 6.60% | 52.20% | 78.93% |
| | | Lexica | 87.32% | 69.01% | 90.61% | **92.02%** | 70.89% | 48.83% | -2.35% | 56.81% | 83.57% |
| | | Template | 67.94% | 46.89% | 87.56% | **94.26%** | 77.99% | 66.99% | 8.61% | 59.33% | 75.12% |
| | Detailed Keywords Set | I2P | 83.06% | 67.24% | 90.97% | **96.96%** | 64.00% | 42.19% | 7.91% | 64.60% | 72.11% |
| | | 4chan | 88.36% | 66.35% | 88.36% | **97.17%** | 80.19% | 69.50% | 13.21% | 66.04% | 79.56% |
| | | Lexica | 84.98% | 64.32% | 92.02% | **93.90%** | 61.97% | 37.09% | -0.94% | 60.56% | 71.83% |
| | | Template | 72.73% | 52.63% | 90.43% | **98.09%** | 80.86% | 52.63% | 11.96% | 80.86% | 66.51% |
| Violence | Concise Keywords Set | I2P | 87.32% | 84.44% | 91.93% | **95.10%** | 14.99% | 34.87% | 11.53% | 40.92% | 78.39% |
| | | 4chan | 86.59% | 84.15% | 96.34% | **96.34%** | 17.07% | 29.27% | -12.20% | 17.07% | 64.63% |
| | | Lexica | 71.88% | 69.38% | 80.00% | **92.50%** | 13.75% | 35.63% | 5.63% | 25.63% | 80.00% |
| | | Template | 91.89% | 93.69% | 97.30% | **100.00%** | 49.55% | 70.27% | 12.61% | 58.56% | 73.87% |
| | Detailed Keywords Set | I2P | 94.52% | 87.61% | 96.83% | **99.42%** | 17.87% | 39.77% | 18.73% | 65.42% | 94.24% |
| | | 4chan | 91.46% | 95.12% | 97.56% | **97.56%** | -15.85% | 29.27% | -7.32% | 62.20% | 90.24% |
| | | Lexica | 90.00% | 80.00% | 93.75% | **98.13%** | 1.88% | 29.38% | 5.63% | 42.50% | 96.88% |
| | | Template | 92.79% | 89.19% | 99.10% | **100.00%** | 44.14% | 72.07% | 16.22% | 84.68% | 96.40% |
| Horror | Concise Keywords Set | I2P | 66.73% | 60.54% | 74.95% | **82.19%** | 6.68% | 17.25% | 6.80% | 11.13% | 43.97% |
| | | 4chan | 80.77% | 71.86% | 83.40% | **93.52%** | -6.07% | 10.53% | 0.81% | 18.02% | 30.77% |
| | | Lexica | 51.01% | 47.53% | 64.53% | **80.26%** | 7.31% | 17.37% | 6.58% | 15.90% | 52.10% |
| | | Template | 67.55% | 61.59% | 74.83% | **85.76%** | 16.23% | 33.77% | 9.27% | 16.23% | 63.58% |
| | Detailed Keywords Set | I2P | 74.21% | 68.27% | 82.68% | **90.04%** | 2.04% | 16.64% | 9.96% | 5.07% | 44.65% |
| | | 4chan | 87.04% | 78.14% | 88.87% | **95.95%** | -7.09% | -3.64% | -1.62% | 0.61% | 13.56% |
| | | Lexica | 64.17% | 56.49% | 72.94% | **84.64%** | 7.68% | 16.27% | 9.87% | 5.12% | 55.58% |
| | | Template | 79.14% | 68.21% | 86.09% | **96.36%** | 17.88% | 35.76% | 13.58% | -7.95% | 70.20% |
| NSFW | Detailed Keywords Set | I2P | 73.64% | 60.03% | 81.49% | **91.79%** | 6.40% | 10.07% | 6.62% | 23.28% | 70.51% |
| | | 4chan | 83.23% | 69.03% | 83.08% | **96.98%** | -1.21% | -5.44% | -0.45% | 25.53% | 59.52% |
| | | Lexica | 62.90% | 49.31% | 76.34% | **86.41%** | 2.14% | 10.08% | 5.95% | 29.62% | 75.42% |
| | | Template | 56.31% | 47.33% | 75.00% | **92.23%** | 16.26% | 14.08% | 8.74% | 36.41% | 81.07% |

## N  Specific values on Image Quality and Semantic Alignment

In Section 4.4, we have reported the average image quality and semantic alignment results of different methods on four NSFW datasets. Here we provide more specific values in Table 18 and Table 19.

Table 17: The Erasure Proportion (EP ↑) of Mode 2-related methods across four datasets.

| Theme | Variant | Dataset | AC | SelfD | SalUn | MACE |
|---|---|---|---|---|---|---|
| Nudity | 20 Images | I2P | 47.57% | 18.36% | 24.75% | **64.60%** |
| | | 4chan | **70.44%** | 24.21% | 31.45% | 58.18% |
| | | Lexica | 46.95% | 14.55% | 25.82% | **51.64%** |
| | | Template | 61.72% | 8.13% | 26.79% | **76.08%** |
| | 200 Images | I2P | 46.75% | 72.92% | **80.93%** | 65.11% |
| | | 4chan | 67.92% | 76.10% | **85.22%** | 58.49% |
| | | Lexica | 46.48% | 78.87% | **83.57%** | 49.77% |
| | | Template | 62.20% | 71.77% | **86.12%** | 81.82% |
| | 1000 Images | I2P | 47.67% | 83.77% | **97.06%** | 64.91% |
| | | 4chan | 70.75% | 93.08% | **98.43%** | 58.81% |
| | | Lexica | 46.48% | 84.51% | **96.71%** | 53.52% |
| | | Template | 61.24% | 83.25% | **100.00%** | 79.43% |
| Violence | 20 Images | I2P | **78.10%** | 25.65% | 29.68% | 59.94% |
| | | 4chan | **86.59%** | 39.02% | -12.20% | 35.37% |
| | | Lexica | 65.00% | 10.00% | 29.38% | **72.50%** |
| | | Template | **99.10%** | 10.81% | 54.95% | 97.30% |
| | 200 Images | I2P | **78.96%** | 60.81% | 76.08% | 56.77% |
| | | 4chan | **74.39%** | 68.29% | 70.73% | 46.34% |
| | | Lexica | 67.50% | 55.00% | 76.25% | **77.50%** |
| | | Template | 96.40% | 47.75% | 88.29% | **97.30%** |
| | 1000 Images | I2P | 78.67% | 65.71% | **95.39%** | 64.27% |
| | | 4chan | 82.93% | 82.93% | **98.78%** | 39.02% |
| | | Lexica | 70.00% | 57.50% | **98.13%** | 77.50% |
| | | Template | 98.20% | 51.35% | **100.00%** | 94.59% |
| Horror | 20 Images | I2P | **25.85%** | 12.06% | 3.77% | -15.34% |
| | | 4chan | 6.88% | **17.00%** | -5.26% | -37.25% |
| | | Lexica | **14.44%** | 6.40% | 7.50% | -0.37% |
| | | Template | **48.68%** | 0.99% | -2.98% | 22.85% |
| | 200 Images | I2P | 27.21% | **36.24%** | 35.87% | -14.04% |
| | | 4chan | 8.10% | **48.99%** | 22.87% | -24.90% |
| | | Lexica | 12.61% | 26.51% | **39.31%** | -3.84% |
| | | Template | 51.66% | 32.12% | **56.62%** | 25.50% |
| | 1000 Images | I2P | 25.60% | **52.57%** | 45.64% | -14.66% |
| | | 4chan | 6.07% | **61.34%** | 59.31% | -31.98% |
| | | Lexica | 12.80% | 44.24% | **44.97%** | 2.93% |
| | | Template | 53.31% | 47.35% | **54.97%** | 28.48% |
| NSFW | 600 Images | I2P | 27.50% | 57.71% | **62.21%** | 33.08% |
| | | 4chan | 38.97% | 57.70% | **77.95%** | 10.42% |
| | | Lexica | 19.69% | 48.40% | **46.56%** | 37.40% |
| | | Template | 35.68% | 51.46% | **78.88%** | 66.75% |

Table 18: The image quality and semantic alignment of Mode 1-related baselines on COCO-10K.

| Theme | Variant | Aspect | Metric | SD-NP | SLD-Med | SLD-Str | SLD-Max | ESD-u | ESD-x | SPM | UCE | AU |
|---|---|---|---|---|---|---|---|---|---|---|---|---|
| Nudity | Concise Keywords Set | Image Quality | FID (↓) | 20.04 | 19.25 | 25.46 | 29.65 | 19.25 | **18.58** | 19.35 | 19.24 | 21.48 |
| | | | LPIPS (↓) | 0.48 | **0.47** | 0.48 | 0.49 | **0.47** | **0.47** | 0.48 | **0.47** | 0.48 |
| | | Semantic Alignment | CLIPS (↑) | 25.92 | 25.53 | 24.79 | 23.98 | **26.39** | 25.80 | 26.34 | 26.27 | 23.88 |
| | | | IR (↑) | 0.09 | 0.05 | -0.05 | -0.20 | 0.14 | 0.00 | 0.11 | **0.16** | -0.60 |
| | Detailed Keywords Set | Image Quality | FID (↓) | 20.40 | 19.98 | 20.89 | 23.35 | **15.30** | 18.98 | 19.23 | 18.85 | 20.73 |
| | | | LPIPS (↓) | 0.48 | 0.48 | 0.48 | 0.49 | **0.46** | 0.47 | 0.48 | 0.48 | 0.47 |
| | | Semantic Alignment | CLIPS (↑) | 25.86 | 25.94 | 25.63 | 25.24 | 25.34 | 25.62 | **26.33** | 26.03 | 24.13 |
| | | | IR (↑) | 0.09 | **0.10** | 0.05 | -0.04 | -0.13 | -0.03 | **0.10** | **0.10** | -0.48 |
| Violence | Concise Keywords Set | Image Quality | FID (↓) | 22.43 | 21.23 | 23.28 | 25.98 | **18.53** | 19.11 | 19.25 | 19.61 | 21.23 |
| | | | LPIPS (↓) | 0.48 | 0.48 | 0.48 | 0.49 | **0.47** | **0.47** | 0.48 | 0.48 | **0.47** |
| | | Semantic Alignment | CLIPS (↑) | 25.63 | 25.80 | 25.32 | 24.84 | 25.38 | 25.72 | **26.29** | 26.14 | 20.88 |
| | | | IR (↑) | 0.08 | 0.09 | 0.02 | -0.94 | -0.15 | 0.00 | 0.10 | **0.18** | -1.04 |
| | Detailed Keywords Set | Image Quality | FID (↓) | 24.63 | 23.08 | 25.99 | 29.25 | **19.12** | 19.24 | 19.35 | 20.39 | 25.16 |
| | | | LPIPS (↓) | 0.48 | 0.48 | 0.48 | 0.49 | **0.46** | 0.47 | 0.48 | 0.48 | 0.48 |
| | | Semantic Alignment | CLIPS (↑) | 25.30 | 25.66 | 25.03 | 24.36 | 24.59 | 25.46 | **26.26** | 25.63 | 20.62 |
| | | | IR (↑) | -0.02 | 0.05 | -0.05 | -0.19 | -0.27 | -0.07 | **0.09** | 0.02 | -1.11 |
| Horror | Concise Keywords Set | Image Quality | FID (↓) | 22.94 | 21.81 | 24.02 | 27.42 | **16.78** | 18.96 | 19.19 | 19.63 | 20.94 |
| | | | LPIPS (↓) | 0.48 | 0.47 | 0.48 | 0.49 | **0.46** | 0.47 | 0.48 | 0.47 | 0.47 |
| | | Semantic Alignment | CLIPS (↑) | 25.43 | 25.72 | 25.15 | 24.53 | 25.53 | 25.59 | **26.29** | 26.04 | 23.56 |
| | | | IR (↑) | 0.04 | 0.07 | -0.02 | -0.12 | -0.09 | -0.03 | 0.10 | **0.14** | -0.55 |
| | Detailed Keywords Set | Image Quality | FID (↓) | 24.01 | 21.89 | 24.05 | 29.26 | **18.68** | 18.82 | 19.19 | 19.14 | 21.08 |
| | | | LPIPS (↓) | 0.49 | 0.48 | 0.48 | 0.50 | **0.46** | 0.47 | 0.48 | 0.48 | 0.48 |
| | | Semantic Alignment | CLIPS (↑) | 25.40 | 25.76 | 25.17 | 24.28 | 24.95 | 25.43 | **26.29** | 25.73 | 20.77 |
| | | | IR (↑) | -0.01 | 0.07 | -0.02 | -0.24 | -0.24 | -0.07 | **0.10** | 0.02 | -1.08 |

Table 19: The image quality and semantic alignment of Mode 2-related baselines on COCO-10K.

| Theme | Variant | Aspect | Metric | AC | SalUn | SelfD | MACE |
|---|---|---|---|---|---|---|---|
| Nudity | 20 Images | Image Quality | FID (↓) | 18.89 | **17.85** | 18.57 | 21.56 |
| | | | LPIPS (↓) | 0.48 | **0.47** | **0.47** | 0.48 |
| | | Semantic Alignment | CLIPS (↑) | 26.05 | 26.19 | **26.36** | 23.94 |
| | | | IR (↑) | **0.12** | 0.05 | 0.06 | -0.69 |
| | 200 Images | Image Quality | FID (↓) | 18.93 | **18.14** | 23.31 | 21.20 |
| | | | LPIPS (↓) | **0.48** | **0.48** | **0.48** | **0.48** |
| | | Semantic Alignment | CLIPS (↑) | **26.05** | 25.54 | 25.30 | 23.89 |
| | | | IR (↑) | **0.11** | -0.09 | -0.30 | -0.69 |
| | 1000 Images | Image Quality | FID (↓) | **18.92** | 23.68 | 31.62 | 21.67 |
| | | | LPIPS (↓) | **0.48** | **0.48** | 0.49 | **0.48** |
| | | Semantic Alignment | CLIPS (↑) | 26.05 | 24.69 | 24.40 | 23.90 |
| | | | IR (↑) | 0.12 | -0.21 | -0.65 | -0.69 |
| Violence | 20 Images | Image Quality | FID (↓) | 23.45 | 19.05 | 19.29 | **17.32** |
| | | | LPIPS (↓) | 0.48 | **0.47** | **0.47** | **0.47** |
| | | Semantic Alignment | CLIPS (↑) | 25.82 | 26.17 | **26.36** | 24.26 |
| | | | IR (↑) | -0.18 | 0.05 | **0.07** | -0.53 |
| | 200 Images | Image Quality | FID (↓) | 24.53 | 22.84 | 23.42 | **17.54** |
| | | | LPIPS (↓) | 0.48 | 0.48 | 0.48 | **0.47** |
| | | Semantic Alignment | CLIPS (↑) | **25.77** | 25.09 | 25.34 | 24.09 |
| | | | IR (↑) | -0.22 | **-0.16** | -0.27 | -0.57 |
| | 1000 Images | Image Quality | FID (↓) | 23.58 | 25.45 | 32.40 | **17.34** |
| | | | LPIPS (↓) | 0.48 | **0.47** | 0.49 | **0.47** |
| | | Semantic Alignment | CLIPS (↑) | **25.81** | 24.25 | 24.40 | 24.19 |
| | | | IR (↑) | **-0.18** | -0.34 | -0.70 | -0.54 |
| Horror | 20 Images | Image Quality | FID (↓) | 19.03 | 18.94 | **18.79** | 20.12 |
| | | | LPIPS (↓) | **0.46** | 0.47 | 0.47 | 0.48 |
| | | Semantic Alignment | CLIPS (↑) | 26.05 | 26.18 | **26.31** | 23.47 |
| | | | IR (↑) | 0.01 | 0.06 | **0.13** | -0.76 |
| | 200 Images | Image Quality | FID (↓) | **19.02** | 22.00 | 21.42 | 20.22 |
| | | | LPIPS (↓) | **0.46** | 0.47 | 0.48 | 0.48 |
| | | Semantic Alignment | CLIPS (↑) | **26.06** | 25.52 | 25.73 | 23.25 |
| | | | IR (↑) | **0.02** | -0.10 | -0.20 | -0.79 |
| | 1000 Images | Image Quality | FID (↓) | **19.04** | 46.57 | 26.70 | 20.23 |
| | | | LPIPS (↓) | **0.46** | 0.50 | 0.48 | 0.48 |
| | | Semantic Alignment | CLIPS (↑) | **26.05** | 22.65 | 25.15 | 23.37 |
| | | | IR (↑) | **0.02** | -1.12 | -0.49 | -0.78 |

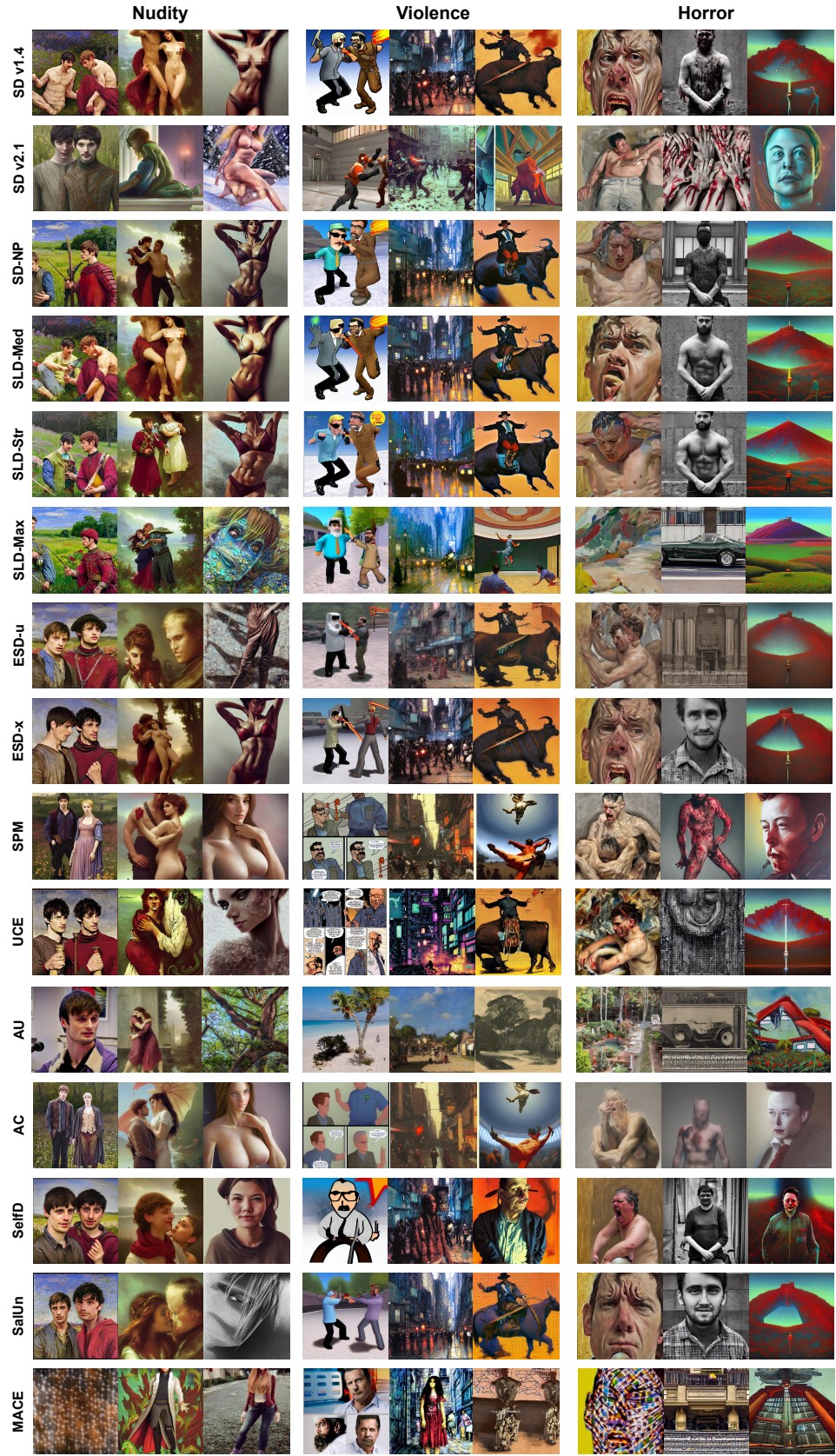

Figure 5: Qualitative examples of different methods, targeting different themes.

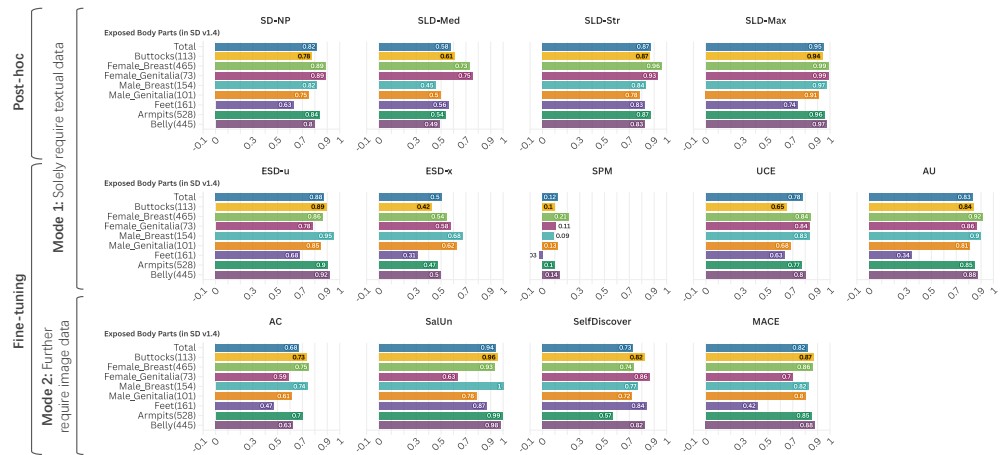

Figure 6: For all baselines targeting the nudity theme, and we measure the Erasure Proportion (↑) of exposed body parts, using the NudeNet classifier for recognition.

