# OpenReview forum: "Comprehensive Assessment and Analysis for NSFW Content Erasure in Text-to-Image Diffusion models"
_NeurIPS.cc/2025/Datasets_and_Benchmarks_Track — NeurIPS 2025 Datasets and Benchmarks Track poster_

### Official Review · Reviewer_cmro · 2025-06-16

**Rating:** 4
**Confidence:** 3

**Summary:**

This paper presents a systematic benchmarking framework for evaluating concept erasure techniques aimed at removing not-safe-for-work (NSFW) content from text-to-image diffusion models.

**Dataset Code Accessibility:**

Yes

**Ethical Considerations:**

No, there are no or only very minor ethics concerns

**Final Justification:**

Thanks for all your justification and I think this paper gives a complete benchmark using diffusion model with open-source code. The authors carefully addressed comments reviewers raised.

**Limitations Weaknesses:**

1. The benchmark focuses only on three NSFW subcategories: nudity, violence, and horror , explicitly excluding political, racial, or legally sensitive categories, which are common in real-world misuse cases (p.9, line 348). This narrow scope limits the benchmark’s coverage and generalization to broader definitions of “unsafe content.” I suggest that future versions could extend thematic coverage by incorporating additional categories such as hate speech, abuse, or misinformation, even if initially limited to small-scale annotations or partially automated detection.


2. While the paper performs fine-grained annotations via expert labeling (p.4, Sec. 3.2), it acknowledges inconsistencies in previous datasets ( I2P) and variability in human interpretation (p.6). However, no inter-annotator agreement metrics are reported, nor are steps taken to minimize subjectivity formally quantified.

3.Although dataset cleaning is listed as one category of erasure (Table 2), the paper excludes these methods from core experiments due to cost and impracticality (p.6, Sec. 4.1). However, this omits an important class of interventions used in commercial systems like DALLE 3 or SD2.1

4. All experiments are conducted using Stable Diffusion v1.4 as the base model (p.6, Sec. 4). This limits the generality of the findings to a single architecture and training dataset, while diffusion models like SD 2.1, DALLE 3, and Midjourney differ significantly in architecture, data curation, and filtering. My major concern is the results may not generalize to newer or commercial models with different inductive biases or pretrained safety filters.


5. Although the authors provide code (p.1), the main paper lacks clear documentation of certain implementation details such as training  time comparisons are deferred to Appendix H. Also, the hyperparameters, batch size and training epochs are not explained. A summary table of method-specific settings and computational requirements should be included in the main paper for transparency and reproducibility.


6.The paper does not include qualitative visual examples of failure cases, such as prompts that bypass erasure filters, unintended side effects. Without failure analysis, it's hard to understand where and why methods fail, especially in borderline or adversarial cases.

**Strengths Contributions:**

1. The paper addresses a critical and underexplored problem in generative AI, which is the systematic evaluation of NSFW concept erasure in text-to-image diffusion models. As public adoption of models like Stable Diffusion grows, so does the need for safety mechanisms. This work contributes a much-needed evaluation standard for developers and researchers seeking to deploy safer generative models (p.1-2).

2. The authors introduce the first comprehensive benchmark specifically tailored to NSFW concept erasure (Table 1, p.2), filling a notable gap left by existing works like UnsafeD [29], UCANVAS [45], and HUB [25], which either focus on base model safety or unrelated concepts such as style or object erasure.

3. The paper categorizes methods by the training stage, required data types, and targeted model components. The paper is clearly written and logically organized, with intuitive sectioning (e.g., Evaluation Framework, Results and Analysis), well-labeled tables (Table 1, Table 3, Table 4, Table 5), and informative figures (Figure 1: Evaluation pipeline; Figure 2: Dataset toxicity relationships).

---

> ### Author Rebuttal · Authors · 2025-07-30
>
> Thank you for the thoughtful and constructive feedback that helped us improve our work. In address to the questions you raised, we have made responses below.
>
> **L1**: Nudity, violence, and horror are representative NSFW themes that cover a substantial portion of unsafe content. Most baseline methods were evaluated only on nudity in their original papers(e.g., SalUn, SelfD, MACE), and even related benchmarks like UCANVAS and HUB do not assess as broad a range of NSFW themes as our study. We exclude hate and abuse content from evaluation because these themes are highly context-dependent and often lack distinctive visual signatures, making detection highly subjective and difficult to standardize without rich textual or contextual information.
>
>
>
> **L2**: As lines 108–111, the human annotations were based on the category definitions provided in Appendix B. In most cases, annotators’ judgments are consistent. For the minority of cases where discrepancies arise (e.g., whether screaming indicates horror), we apply majority voting among annotators to minimize subjectivity, which is then formally quantified.
>
> **L3**: We present the performance of SD 2.1 on NSFW content in Tab. 11 (App. G). To include it in core experiments and evaluate its behavior on specific themes, one would need to remove corresponding concept-related images from its original training data. However, as the native training dataset for SD 2.1 and Dalle is not publicly available, such experiments cannot be conducted. In real-world scenarios, cleaning training datasets also incurs high human effort. This motivates the development of concept erasure methods that solve this problem at a lower cost. These concept erasure methods are the primary focus of our evaluation.
>
> **L4**: Both DALL·E 3 and Midjourney are closed-source models, which prevents us from accessing their code, let alone performing training. To tackle this, we carry out our experiments on a similar and open-source diffusion model—Stable Diffusion.
>
> **L5**: For training, as lines 221–222, we follow the configurations and hyperparameters suggested in the original papers. For generating, each baseline diffuses 40 steps. To improve clarity, we have explicitly included method-specific settings and computational requirements as following table:
>
> | **Methods** | **Lr** | **Steps** | **Other hyperparameters** |
> | --- | --- | --- | --- |
> | SD-NP | / | / | guidance scale=7.5 |
> | SLD-Med | / | / | sld warmup steps=10; sld guidance scale=1000; sld threshold=0.01; sld momentum scale=0.3; sld mom beta=0.4 |
> | SLD-Str | / | / | sld warmup steps=7; sld guidance scale=2000; sld threshold=0.025; sld momentum scale=0.5; sld mom beta=0.7 |
> | ESD-u | 1e-5 | 1000 | start guidance=3; negative guidance=1 |
> | ESD-x | 1e-5 | 1000 | start guidance=3; negative guidance=1 |
> | SPM | 1e-4 | 3000 | lr warmup steps=500; text encoder lr=5e-5; guidance scale=3 |
> | UCE | / | / | erase scale=1; guidance scale=7.5 |
> | AU | 1e-5 | 1000 | save interval=200; retain step=1; retain loss w=1; attack lr=1e-3; attack step=30; warmup iter=200; negative guidance=1 |
> | AC | 2e-6 | 200 | / |
> | SelfD | 1e-1 | 20 | lr scheduler=constant |
> | SalUn | 1e-5 | 1000 | alpha=0.1; mask threshold=0.5; guidance scale=7.5 |
> | MACE | 1e-5 | 50 | negative guidance=1.0; uncond loss weight=1 |
>
>
> **L6**:  We initially reduced the display of unsafe images in the main paper out of consideration for reviewer comfort. However, we acknowledge that additional qualitative comparisons would help better illustrate differences across baselines. We have included more qualitative results in the revised version.
>
> ---
>
> Thank you again for your valuable feedback! We believe that this submission has been further improved based on your suggestions. We hope these responses can address all your concerns and convince you to lean more towards acceptance of our paper.

---

> ### Author Response · Authors · 2025-08-06
>
> Dear Reviewers cmro,
>
> We sincerely appreciate your time in providing detailed comments and helpful suggestions! We believe that this submission has been further improved according to your helpful suggestions.
>
> As the due of the discussion phase is very close, we are wondering if our response meets your expectations. We are eager to engage in further discussions and continue improving our work. Thank you again!
>
> Authors

---

> ### Author Response · Authors · 2025-08-08
>
> Dear Reviewers cmro,
>
> We sincerely thank you for your time and insightful comments. In response to suggestions from you and other reviewers, we have carefully revised and improved the manuscript.
>
> The key updates include:
> + Enhanced qualitative comparisons: We have added more comprehensive qualitative results in the revised version to better illustrate the performance differences across methods.
> + Additional experimental evaluations: We have conducted further experiments with additional baseline methods, including ACE, EraseDiff, and Meta-Unlearning.
> + Improved table presentation: The header of Table 1 has been updated for greater clarity and consistency.
> + Inference time included: We now report the inference time for all baseline methods, providing a complete table of computational efficiency.
> + Analysis of context-dependent failure modes: We have clarified the underlying causes of context-dependent performance degradation (referred to as "context-dependent horror") and provided explanations for the limitations of fine-tuning-based approaches. Furthermore, we propose potential solutions to mitigate these issues.
> + Application recommendations: We have added practical recommendations for selecting the most suitable concept erasure method depending on different real-world application scenarios.
> + Explicit rationale for content exclusion: We now clearly state and justify in the manuscript why certain categories—such as hate, extremism, abuse, and misinformation—are excluded from our study.
> + Method-specific settings and computational requirements: Detailed implementation settings and computational resource requirements for each method have been explicitly included in the supplementary tables.
>
> We believe these revisions have significantly strengthened the paper. If you have any further questions or suggestions, we would be happy to address them.
>
> Authors

---

### Official Review · Reviewer_YDju · 2025-07-02

**Ethics Flags:** Data privacy, copyright, and consent,…
**Rating:** 5
**Confidence:** 3

**Summary:**

The paper introduces the first end-to-end benchmark and toolkit for studying NSFW-concept erasure in text-to-image diffusion models. It builds a pipeline that spans (i) five annotated datasets, (ii) a refined taxonomy covering 13 state-of-the-art erasure methods, (iii) automatic detectors, and (iv) metrics that jointly assess erasure success, image quality, semantic alignment and robustness. Extensive experiments on Stable-Diffusion-v1.4 compare post-hoc, parameter fine-tuning and data-cleaning approaches under four evaluation perspectives.

**Additional Feedback:**

It would be great if author can try to address the limitation and ethical review. As for the easy fix for the human evaluation. I suggest author either provide a small cohort study to experiment 5.4. Or just add more example images.

**Dataset Code Accessibility:**

Yes

**Dataset Code Comments:**

Code is well structured with clear datasets, evaluate, method. I trust the author on reproducibility.

**Ethical Comments:**

Safety and security
Benchmarked methods claim to remove NSFW content, but as the paper notes, adversarial prompts can circumvent many approaches. There's risk that tools may give false assurances of safety and be deployed where restricted content is undesirable (e.g., education, workplace).

Authors are encouraged to implement stronger threat models, including adaptive white-box attacks, and clearly report failure modes. They should also state caution for deployment contexts, recommending monitoring instead of blind trust.

**Ethical Considerations:**

Yes, there are ethics concerns that require attention by the authors

**Final Justification:**

This work offers a timely, comparatively comprehensive benchmark for NSFW concept erasure in diffusion models, with open code and added experiments addressing key reviewer concerns. Given the scope, careful experimental design, and utility to the community, I would like to recommend acceptance.

**Limitations Weaknesses:**

- I feel like it's asking too much but why all those entensive experiment are just based a single backbone :SD-v1.4. What's the rational of not using SD-XL or DALL·E 3?
- NSFW themes is limited to nudity, violence, horror. However, hate, extremist, self-harm content are omitted.
- One more thing is there is missing human evaluation. No user study on perceptual quality or residual NSFW; solely automated metrics may induce bias.
- Latest 2025 methods (EraseDiff, TRACE, Meta-Unlearning) absent from results.

**Strengths Contributions:**

- Prior work evaluates single methods or unsafe-image benchmarks (e.g. UnsafeD, HUB) but no comprehensive comparison exists for NSFW erasure; the paper fills this gap.
- Fine-grained annotations for nudity, violence, horror, enabling nuanced analysis.
- Extensive experiments – 13 SOTA baselines trained on SD-1.4 with open-sourced code; >80 k generations on A100 GPUs; training-time and hyperparam grids reported.
- Paper is easy to follow; tables/figures are informative
- The insight provided for each experiments are quite interesting.

---

> ### Author Rebuttal · Authors · 2025-07-30
>
> Thank you for the thoughtful and constructive feedback that helped us improve our work. In address to the questions you raised, we have made responses below.
>
> **L1**: Baseline methods in the original papers were implemented using SD 1.4, due to its lower training cost. Besides, the choice of the base model does not inherently affect the effectiveness of the erasure methods themselves, as these base models are all based on the diffusion process and share similar modular designs. To ensure consistency across comparisons, we adopt a uniform backbone (SD 1.4) in all experiments.
>
> **L2**: Nudity, violence, and horror are representative NSFW themes that cover a substantial portion of unsafe content. Most baseline methods were evaluated only on nudity in their original papers(e.g., SalUn, SelfD, MACE), and even related benchmarks like UCANVAS and HUB do not assess as broad a range of NSFW themes as our study. Compared to prior work, our benchmark represents a significant step forward in comprehensiveness. Self-harm, as mentioned, is grouped under violence in our benchmark due to strong visual similarities with violent content. We exclude hate and extremist content from evaluation because these themes are highly context-dependent and often lack distinctive visual signatures, making detection highly subjective and difficult to standardize without rich textual or contextual information.
>
> **L3/Additional Feedback**: When selecting automated classifiers, we conducted human annotation as ground truth to identify those well-aligned with human judgments, ensuring reliable automatic evaluation in subsequent experiments. We initially reduced the display of unsafe images in the main paper out of consideration for reviewer comfort. However, we acknowledge that additional qualitative comparisons would help better illustrate differences across baselines. We have included more qualitative results in the revised version.
>
> **L4**:  Thank you for pointing out the new related work. We have conducted additional experimental results on EraseDiff and Meta-Unlearning, which will be incorporated into the latest version. To the best of our knowledge, there is no existing TRACE-related work that directly intersects with our research area. The results are summarized below:
> |  |  | Erasure Effect | | | | | Image Quality | | Semantic Alignment | |
> | --- | --- | --- | --- | --- | --- | --- | --- | --- |--- |--- |
> | Method | Theme | I2P (EP ↑) | 4chan (EP ↑) | Lexica (EP ↑) | Template (EP ↑) | Overall (EP ↑) | FID (↓) | LPIPS (↓) | CLIPS (↑) | IR (↑) |
> | EraseDiff | NSFW | 19.60% | 11.78% | 13.13% | 19.17% | 17.16% | 20.14 | 0.47 | 25.86 | -0.00068 |
> | Meta-Unlearning | NSFW | 5.17% | -1.36% |-1.53% | 1.21% | 2.54% | 18.91 | 0.47 | 25.86 | 0.11|
>
> ---
>
>
> **Ethical Comments**：We fully acknowledge the ethical concerns surrounding the potential misuse or misinterpretation of NSFW concept erasure methods. These concerns directly motivate our work, which offers the first comprehensive evaluation of their effectiveness—crucial for identifying potential risks and encouraging safe, informed deployment. Our study also provides practical guidance for responsible application in real-world settings, aiming to advance understanding of content safety in diffusion models and support future research in this important area. We sincerely appreciate your suggestion on incorporating more advanced adaptive attack strategies and explicit deployment cautions, which are highly valuable for refining our study and guiding future work.
>
> ---
>
> Thank you again for your valuable feedback! We believe that this submission has been further improved based on your suggestions. We hope these responses can address all your concerns and convince you to lean more towards acceptance of our paper.

---

> > ### Comment · Reviewer_YDju · 2025-08-01
> >
> > Thanks for the responses and added results (EraseDiff/Meta-Unlearning).
> >
> > I only have one point to add regarding the scope of NSFW content:
> > If hate/extremism are excluded, state this more prominently and add rationale in the manuscript. For self-harm folded into violence, provide the label mapping and examples so readers understand edge cases.

---

> > ### Author Response · Authors · 2025-08-02
> >
> > We are pleased to receive your feedback, and thank you for your efforts and time. We have clearly stated the scope and rationale for excluding hate and extremist content in the manuscript. Additionally, we have added label mapping and illustrative examples for self-harm to clarify how such cases are handled and to improve transparency for readers. While we have excluded certain themes, we have made our benchmark more comprehensive and systematic than existing works. We hope these contributions will help advance the field and inspire future research toward more robust and transparent practices.

---

### Official Review · Reviewer_yeGd · 2025-07-19

**Rating:** 5
**Confidence:** 3

**Summary:**

This work builds a comprehensive benchmark of concept erasure methods for diffusion models in text-to-image NSFW scenarios, which involves 13 concept erase methods and NSFW classifiers for evaluation. Moreover, the authors investigate the NSFW themes, data scales, and robustness to reveal the difference in the methods' capability.

**Additional Feedback:**

I acknowledge the efforts and values of this work for assessing the concept erasure methods for text-to-image safety issues. However, the position or the guide is partly unclear to me, as mentioned in **Limitations Weaknesses**. Please address my unclear parts and correct if there is any misunderstanding.
In addition (optionally), if you can give concrete insights and guidelines to readers, it would be better.

**Dataset Code Accessibility:**

Partly

**Dataset Code Comments:**

In the first step, "1. Place the trained model weights in the `models` directory. Refer to `method/def.py` for detailed specifications."
Could you provide how to access the model weights (https://github.com/OPTML-Group/Unlearn-Saliency) you mentioned in the file?

**Ethical Considerations:**

No, there are no or only very minor ethics concerns

**Final Justification:**

There is no critical question from my side.
I am pleased to see that the authors actively response to other reviews' comments and widen their coverage of this benchmark.
I believe this benchmark would help practitioners study more erase methods for diffusion models for AI safety.

**Limitations Weaknesses:**

While the authors reveal their benchmark results and analysis in detail, there are some unclear parts I would like to check.

1. Table 1 could be consistent with Section 4. For example, change the order of the column of **Taxonomy of Erasure Methods** and **Multi Themes**; group **Multi Themes** and **Data Scales** as the **Sensitivity to Training Data**.
2. In Sec 3.4, do you use fine-tuned NSFW-related classifiers for evaluating the discrepancies between human and model judgments? Because VQA could accurately capture different NSFW themes, could you provide which VQA model for image-text alignment scoring?
3. Following 2, in Lines 185-186, 'MHSC tends to be conservative, while VQA is more sensitive.' How do we get this conclusion? Do I miss something in the main manuscript?
4. In Lines 252-257 and Appendix H, could you provide the inference time besides comparing the training time? It is valuable to know that a concept earsure method's inference time when end-users use this diffusion model.
5. In Lines 247-248, 'This could be attributed to the fact that fine-tuning-based methods rely on concept-specific hyperparameters'. What are the *concept-specific hyperparameters*? Could you give an example? What are the sensitive hyperparameters that affect the benchmark results?
6. In Lines 273-276, '... fine-tuning-based methods perform poorly on the horror theme compared to other themes' and '... horror is a more abstract and context-dependent concept, often conveyed through mood, atmosphere, and symbolic imagery.' Could you explain the causation of context-dependent horror and the failure of poor performance of fine-tuning-based methods? What's the challenge of using a fine-tune-based method for horror-theme datasets?
7. In Line 288 ***Takeaways***, 'Blindly increasing the training data scale might offer limited benefits for most concept erasure methods'. How could we conclude it based on the previous paragraphs?
8. Following 7, '... while avoiding the need for image dataset collection and mitigating performance risks from low-quality data.' I could not follow the context that there are risks from low-quality data.

The second problem is that, although the authors claim, they provide in-depth insights and practical guidance (guidelines?) for effectively applying concept erasure methods in various real-world scenarios. As a newbie in this field, I cannot easily follow the guide to select the concept earasure methods for a specific scenario. Please try to summarize your claims and results in Sec. 4 as a Table, which conveys the information about which method (row) is suitable (baseline) for which scenarios (column).

**Strengths Contributions:**

1. This work provides a full-pipeline toolkit that allows practitioners can easily start their experiments.
2. They reveal the weak correlation between prompt toxicity and unsafe images, which emphasizes the importance of studying concept erasure techniques.
3. Beyond benchmarking existing concept removal methods for NSFW scenarios, the authors further evaluate the target themes, data scales, robustness, and the impact on the pre-trained diffusion model.

---

> ### Author Rebuttal · Authors · 2025-07-30
>
> Thank you for the thoughtful and constructive feedback that helped us improve our work. In address to the questions you raised, we have made responses below.
>
> **L1**: The header of Tab. 1 has been updated accordingly: : [Benchmark, Key Evaluation Scope, Methods Involved, Taxonomy of Erasure Methods, Erasure Effectiveness, Sensitivity to Training Data, Robustness, Irrelevant Concept Preservation].
>
> **L2**: VQA refers to an existing visual question answering model. We use it directly without any fine-tuning. A more detailed explanation is provided in App. E.
>
> **L3**: In Tab. 9 (App. E), we report the performance of classifiers using accuracy, recall, and F1 score. The recall results support our conclusion: low recall indicates that many images with a ground-truth label of 1 are misclassified as 0, revealing low sensitivity (e.g., MHSC); in contrast, high recall indicates higher sensitivity (e.g., VQA).
>
>
> **L4**：Thank you for highlighting the importance of inference time. We have now included this information below:
>
> |  | SD-NP | SLD-Med | SLD-Str | SLD-Max | ESD-u | ESD-x | SPM | UCE | AU | AC | SelfD | SalUn | MACE |
> | --- | --- | --- | --- | --- | --- | --- | --- | --- | --- | --- | --- | --- | --- |
> | Time (s) | 2.749 | 3.685 | 3.745 | 3.793 | 2.675 | 2.777 | 2.616 | 2.701 | 2.668 | 2.712 | 2.701 | 2.632 | 2.807 |
>
>
> **L5**: Concept-specific hyperparameters refer to the best-performing hyperparameter settings (e.g., fine-tuned modules, learning rate, number of steps) for a specific concept under each fine-tuning method. These require extensive ablation studies to determine. To ensure fair comparison, we adopted the officially recommended general settings provided by the original papers for all baselines.
>
>
> **L6**: Context-dependent concepts like “horror” often require specific combinations of terms in prompts (e.g., "dark" in a horror context) and are reflected in the overall visual tone of the image rather than in localized regions. As mentioned in **L5**, general hyperparameter settings may not yield optimal erasure effects for such abstract concepts. To improve performance in these cases, in addition to searching for optimal settings, one could (i) find the optimal hyperparameter settings, (ii) provide more detailed textual descriptions for text-only methods, or (iii) curate more diverse and representative image datasets for methods that require images.
>
>
> **L7**: This conclusion is drawn from Fig. 3, where lines of different colors represent different training data scales for each method. We observe that for most methods, the lines are nearly overlapping, indicating consistent performance across scales.
>
>
>
> **L8**: Low-quality data can impair the effectiveness of erasure methods, which often require image data for fine-tuning. If the image dataset exhibits fairness issues (e.g., biases in gender or ethnicity) or lacks diversity (e.g., visually similar images), the effectiveness of such methods may be limited to specific or narrow scenarios. In contrast, text-only methods avoid these risks and requirements, making them more broadly applicable.
>
>
>
> **Application Recommendations**: Following your suggestion, we have compiled a recommendation table for concept erasure methods across various real-world scenarios, as shown below. In Tab. 10 (App. F), we also provide a comprehensive and fine-grained comparison of these methods.
>
> | **Scenarios** | **Time-Efficiency Priority** | **Data-Scarce** | **High-Stakes** | **Quality Priority** |
> | --- | --- | --- | --- | --- |
> | Description | Scenarios where fine-tuning is avoided, and inference is performed directly. | Scenarios with low-quality training images or insufficient training data. | Scenarios with prompts posing a high risk of generating harmful images. | Scenarios prioritizing high-quality image generation. |
> | Applicable Methods | SLD | SLD, AU | AU, MACE | ESD-u, SPM |
>
>
> **Dataset Code Comments**: Thank you for your feedback. We will provide the weights of all baseline models we trained via a publicly accessible link, which will be displayed to users as a terminal prompt during inference. Furthermore, we would like to continually improve the automated download and setup pipeline to enhance usability and streamline deployment.
>
> ---
>
> Thank you again for your valuable feedback! We believe that this submission has been further improved based on your suggestions. We hope these responses can address all your concerns and convince you to lean more towards acceptance of our paper.

---

> > ### Comment · Reviewer_yeGd · 2025-08-05
> >
> > Thank you for your clarification with detailed information to guide me in following your manuscript.
> > There is no critical question from my side.
> > I am pleased to see that the authors have added more SOTA methods to expand the coverage of this benchmark.
> > I believe this benchmark would help practitioners study more erase methods for diffusion models for AI safety.
> > Therefore, I will increase my score. Thanks.

---

> > > ### Author Response · Authors · 2025-08-06
> > >
> > > Dear Reviewer yeGd,
> > >
> > > We are grateful for your feedback and are pleased to hear that you no longer have questions. Thank you again for the time and effort you put into reviewing our paper.
> > >
> > > Best regards,
> > >
> > > Authors

---

### Official Review · Reviewer_3Vr5 · 2025-07-23

**Rating:** 5
**Confidence:** 4

**Summary:**

This paper presents a comprehensive toolkit and evaluation benchmark for NSFW concept erasure in text-to-image diffusion models. The authors develop a unified framework that covers dataset construction, a taxonomy of concept erasure methods, high-accuracy evaluation classifiers, and a suite of erasure effectiveness and generative quality metrics. Thirteen state-of-the-art NSFW concept erasure methods are systematically benchmarked under multiple scenarios, including erasure effectiveness, robustness to toxic prompts, sensitivity to data scale, and preservation of unrelated concepts. Extensive empirical results yield in-depth insights into the strengths and limitations of different approaches, and practical guidance is provided for real-world application. The code and toolkit are made publicly available to foster reproducibility and further research.

**Dataset Code Accessibility:**

Yes

**Ethical Considerations:**

No, there are no or only very minor ethics concerns

**Final Justification:**

This work is well-motivated and technically solid. My concern has been solved. I recommend its acceptance. Good Job!

**Limitations Weaknesses:**

1. The authors propose four distinct Evaluation Perspectives: Erasure Effect to Target Concepts, Sensitivity to Training Data Scale, Robustness against Toxic Prompts, and Preservation of Irrelevant Concepts. However, I find the categorization of these perspectives somewhat problematic as they do not reside on the same logical level. For example, Erasure Effect to Target Concepts and Robustness against Toxic Prompts are both fundamentally assessing different facets of the core erasure effectiveness. In contrast, Preservation of Irrelevant Concepts measures an orthogonal aspect: the utility preservation or fidelity of the model concerning concepts unrelated to the erased NSFW content.

2. A small concern is, well, the evaluation heavily relies on automated classifiers with human annotations as ground truth. Section 3.4 mentions varying tolerance levels and label ambiguities, but more analysis and discussion of classifier fairness, calibration, and cross-cultural/subjective differences in NSFW judgment would strengthen trust in the results. Are there cases where automated classifiers strongly disagree with human consensus, and how is this handled?

3. Although a variety of metrics are proposed (EP, GRD, FID, LPIPS, CLIPS, IR), most results are presented numerically. Limited qualitative visual analysis or ablation on how specific metric choices may drive conclusions is included.

4. A highly relevant and recent work is missing: ACE: Concept Editing in Diffusion Models without Performance Degradation.

5. Although Table 4 and Section 4.3 consider adversarial prompt robustness using RAB-generated prompts, concrete illustrations of real-world circumvention or failure scenarios are not deeply explored. But this is not a concern; rather, it is more like a suggestion.

**Strengths Contributions:**

1. The paper addresses an important gap in the literature by offering the systematic and comprehensive evaluation of NSFW concept erasure methods specific to text-to-image diffusion models. The detailed comparison across 13 methods fills a major benchmarking void for practitioners and researchers.

2. The dataset construction process is rigorous, involving fine-grained thematic annotation for multiple NSFW themes across four publicly derived datasets, as shown in Figure 2. This enhances both the depth and breadth of the evaluation, making the benchmark valuable for multi-faceted content safety analyses.

3. The proposed taxonomy organizes concept erasure methods by data requirements and intervention stage, providing clear distinctions that help the community navigate the design space.

---

> ### Author Rebuttal · Authors · 2025-07-30
>
> Thank you for the thoughtful and constructive feedback that helped us improve our work. In address to the questions you raised, we have made responses below.
>
>
> **L1**: You raise a valid point. Our evaluation can indeed be structured around two key levels: erasure effectiveness and preservation of irrelevant concepts. To provide a more fine-grained analysis of the first level, we further break it down into three sub-perspectives, each reflecting a distinct capability:
>
> + Erasure Effect to Target Concepts (Sec. 4.1), comparing overall performance across methods.
> + Sensitivity to Training Data Scale (Sec. 4.2), analyzing each method under different training conditions.
> + Robustness against Toxic Prompts (Sec. 4.3), assessing robustness across methods under adversarial inputs.
>
> Thank you for your feedback. We have incorporated the clarification into the paper to improve its clarity.
>
> **L2**: To highlight potential subjectivity in automated classifiers, Tab. 9 (App. E) compares their performance using human annotations as ground truth, reporting Accuracy, Recall, and F1-score. Notably, the low Recall values indicate that many instances with a true label of 1 are incorrectly classified as 0. For instance, CLIP achieves only 34.95% Recall on the Violence theme, and MHSC achieves just 16.04% on Horror, both of which suggest a significant divergence from human consensus. To handle this issue, we selected VQA with strong performance across all metrics to maximize alignment with human judgments.
>
>
>
> **L3**: We initially reduced the display of unsafe images in the main paper out of consideration for reviewer comfort. However, we acknowledge that additional qualitative comparisons would help better illustrate differences across baselines. We have included more qualitative results in the revised version.
>
>
>
> **L4**: Thank you for pointing out the new related work. We have conducted additional experimental results on ACE. The results are shown in the table below, which will be incorporated into the latest version. ACE shows relatively weak erasure effectiveness, but it can still preserve image quality and semantic content.
>
> |  |  | Erasure Effect | | | | | Image Quality | | Semantic Alignment | |
> | --- | --- | --- | --- | --- | --- | --- | --- | --- |--- |--- |
> | Method | Theme | I2P (EP ↑) | 4chan (EP ↑) | Lexica (EP ↑) | Template (EP ↑) | Overall (EP ↑) | FID (↓) | LPIPS (↓) | CLIPS (↑) | IR (↑) |
> | ACE | NSFW | 17.70% | -0.60% | 13.74% | 27.43% | 14.98% | 20.81 | 0.48 | 25.90 | -0.02 |
>
> ---
>
> Thank you again for your valuable feedback! We believe that this submission has been further improved based on your suggestions. We hope these responses can address all your concerns and convince you to lean more towards acceptance of our paper.

---

### Note · Authors · 2025-08-14

Dear Area Chair and Reviewers,

We sincerely thank you all for your time and dedication throughout the review process. We are grateful to the reviewers for your insightful comments, and to the Area Chair for skillfully overseeing the process with fairness and care.
For your convenience, we briefly summarize our key actions and outcomes below:

1.Enhanced Comparative Analysis

+ We have added more comprehensive qualitative results to better illustrate the performance differences across methods. (Reviewer 3Vr5,YDju,cmro)
+ We have conducted further experiments with additional baseline methods (ACE, EraseDiff, and Meta-Unlearning). (Reviewer 3Vr5,YDju)
+ We now report the inference time for all baseline methods, providing a complete table of computational efficiency. (Reviewer yeGd)

2.Improved Clarity and Explanations
+ The header of Table 1 has been updated for greater consistency. (Reviewer yeGd)
+ We have clarified the causes of context-dependent performance degradation, explained the limitations of fine-tuning-based methods, and proposed potential mitigation strategies.  (Reviewer yeGd)
+ We now clearly state and justify in the manuscript why certain categories are excluded from our study.  (Reviewer YDju,cmro)

3.Enriched Practical Guidance
+ We have added practical recommendations for selecting the most suitable concept erasure method depending on different real-world application scenarios. (Reviewer yeGd)
+ Detailed implementation settings and computational resource requirements for each method have been explicitly included in the supplementary tables. (Reviewer cmro)

**Summary of Reviewer Feedback**
+ Reviewer YDju confirmed that most of their concerns have been adequately addressed.
+ Reviewer yeGd acknowledged the value of our work, stating: “I believe this benchmark would help practitioners” and indicated the intention to increase their scores.
+ Reviewer 3Vr5 provided a positive feedback.
+ We try our best to providing detailed responses to correct all the misunderstandings and to answer all the raised questions. Hope these responses can convince the reviewer cmro to lean more towards acceptance of our paper.

We hope this summary provides a clear reflects the efforts we've made to improve the paper in response to your feedback. Thank you once again to the entire review team for your expertise and support in helping us strengthen this work. We truly appreciate your time and consideration.

---

### Decision · Program_Chairs · 2025-09-18

**Decision:**

Accept (poster)

**Comment:**

This paper introduces the first comprehensive benchmark and toolkit for NSFW concept erasure in text-to-image diffusion models. The reviewers reached a positive consensus (3x accept with either 3/5 or 4/5 confidence; 1 borderline accept with 3/5 confidence). The reviewer who rated the paper as a borderline accept was unresponsive through the rebuttal period, and so I have discounted their review compared to the other reviews.

Overall, the reviewers cited the systematic evaluation of 13 methods, rigorous dataset construction, and practical guidance for real-world use. They also valued the inclusion of human (rather than AI-generated) ground-truth annotations, which were used to validate the automated NSFW classifiers.

The main limitations of the paper were that (i) the dataset/benchmark only covers "three NSFW subcategories: nudity, violence, and horror" and (ii) all the experiments were conducted on Stable Diffusion v1.4 - which is largely considered an 'out-of-date' model. Towards (i), I and the other reviewers were satisfied with the authors' response that these three subcategories cover a substantial portion of unsafe content and are important to detect/erase in and of themselves. Towards (ii), the authors justified their choice of model on cost and reproducibility grounds, which I and the reviewers found reasonable. Despite these limitations, the paper's contributions remain broadly relevant and impactful. The ethics review also confirmed no issues. For the camera-ready, I recommend the authors (i) add more qualitative examples and failure cases, and (ii) include a discussion on applicability to newer models and broader unsafe content categories.

I recommend acceptance as a poster as I believe this will serve as a valuable resource for communities working on content safety.